

# Effects of Ozone-Climate Interactions on the Temperature Variation in the Arctic Stratosphere

Siyi Zhao[1], Jiankai Zhang[1*], Chongyang Zhang[1], Zhe Wang[1]

[1]School of Atmospheric Sciences, Lanzhou University, Lanzhou 730000, China

*Correspondence to*: Jiankai Zhang  (jkzhang@lzu.edu.cn)

**Abstract.** Using reanalysis datasets and the Community Earth System Model (CESM), this study investigates the effects of ozone-climate interactions on the Arctic stratospheric temperature during winter and early spring. From 1980–1999, the Arctic stratospheric temperature increase significantly in early winter (November and December), which is primarily due to ozone-climate interactions. Specifically, the increasing trend in ozone during this period leads to longwave radiation cooling

in the stratosphere. Meanwhile, ozone-climate interactions lead to a stratospheric state that enhances upward wave propagation and the downwelling branch of the Brewer-Dobson circulation, which in turn adiabatically warms the stratosphere and offsets the direct longwave radiative cooling of the ozone. Additionally, enhanced upward wave propagation can lead to an equatorward shifting of the stratospheric polar vortex toward the eastern coast of Eurasia, accompanied by zonally asymmetric anomalies in stratospheric temperature. In contrast, during late winter and spring,

cooling trends in the Arctic stratosphere predominantly driven by the enhanced shortwave radiative cooling associated with stratospheric ozone depletion. After 2000, the response of the Arctic stratospheric temperature trend to ozone changes is weaker than that from 1980–1999. This study highlights the impacts of ozone-climate interactions on intraseasonal variability in the Arctic stratospheric temperature.

**Keywords**: stratospheric temperature, ozone-climate interactions, ozone-circulation feedback, stratospheric polar vortex, planetary wave activity

## 1 Introduction

The stratospheric ozone layer plays an important role in global climate change (Son et al., 2008; Smith and Polvani, 2014; Xia et al., 2016; Xie et al., 2018; Hu et al., 2019a; Sigmond and Fyfe, 2014; Ivanciu et al., 2022). Its absorption of solar

ultraviolet (UV) radiation, along with its strong infrared radiation (IR) absorption and emission at the 9.6 μm band, is crucial for the Earth's energy balance and the thermal structure of the atmosphere. The annual global mean radiative forcing of stratospheric ozone during the strongest ozone depletion period, 1979–1996, is -0.22±0.03 W/m$^2$ (de F. Forster and Shine, 1997). This value is relatively small in comparison to the radiative forcing caused by $CO_2$ (2.16±0.25 W/m$^2$; IPCC, AR5, 2014). However, stratospheric ozone can significantly impact stratospheric temperature variability through its ozone-climate





interactions, which involve a chemical-radiative-dynamical coupling process (Dietmüller et al., 2014). Nowack et al. (2015) demonstrated that neglecting interactive stratospheric chemistry and considering only ozone's direct radiative effect result in a positive surface temperature bias of approximately 20% in the experiment with the quadrupled (4×CO2) experiment. In addition to temperature, previous studies also suggested that climate models without ozone-climate interactions cannot capture the realistic variability in stratospheric compositions and other stratospheric processes (Cionni et al., 2011; Eyring et

al., 2013; Jones et al., 2011). However, Marsh et al. (2016) indicated that ozone variation can only induce a small chemical-radiative-dynamical feedback coefficient (only 1%), suggesting that ozone feedback might not be crucial in modulating stratospheric temperature. It remains unclear how the ozone-climate interactions significantly affect stratospheric climate variability.

The ozone-climate interactions are complex chemical-radiative-dynamical coupling processes associated with ozone changes, which can be described as follows: stratospheric ozone changes affect stratospheric temperature and then influence stratospheric circulation through the adjustment of the thermal wind balance (WMO, 2022). This stratospheric circulation can further influence the dynamic transport of ozone and other stratospheric chemical compositions, and their associated chemical reactions. Specifically, in winter, when solar radiation is absent in the Arctic region, an increase in stratospheric

ozone tends to cool the Arctic lower stratosphere through longwave radiative cooling. This cooling strengthens stratospheric westerlies and the polar vortex, further impeding the transport of ozone-rich air from the mid-latitudes to the Arctic (Strahan et al., 2013). On the other hand, ozone-induced changes in the wave refractive index increase the upward propagation of planetary waves (Haynes et al., 1991; Nathan and Cordero, 2007; Albers and Nathan, 2013; Hu et al., 2015), leading to an increase in the polar temperature and a weakening of the polar vortex. In boreal late winter and early spring, as solar

radiation reaches high latitudes and planetary wave activity weakens, a decrease in the lower stratospheric ozone concentration at the poles can cool the polar regions via absorbing less shortwave radiation. This leads to strengthened westerlies and a stronger stratospheric polar vortex (Coy et al., 1997; Friedel et al., 2022), which in turn leads to less ozone being transported to the poles (Waugh et al., 1997; Hu et al., 2023). Additionally, the low temperature in the Arctic stratosphere facilitates the formation of polar stratospheric clouds (PSCs), on which chlorine reservoirs can convert to active

chlorine (Solomon et al., 1986; Feng et al., 2005a; 2005b). Active chlorine then reacts catalytically with ozone, resulting in ozone depletion. This depletion further cools the Arctic stratosphere by absorbing less UV radiation (Calvo et al., 2015). Therefore, the impact of ozone on stratospheric temperature in winter and spring involves different and complex chemical-radiative-dynamical feedback processes (Tian et al., 2023).

In addition to the stratospheric pathway mentioned above, the stratospheric polar vortex influenced by ozone-climate interactions can also influence tropospheric circulation through the downward control theory (Haynes et al., 1991), wave-zonal flow interactions (Holton and Mass 1976; Christiansen 1999), the rearrangement of stratospheric potential vorticity (Hartley et al., 1998; Black, 2002) and planetary wave propagation induced transient eddy feedback in the upper troposphere



(Zhang et al., 2020; Limpasuvan and Hartmann, 2000; Rao and Garfinkel, 2020, 2021). Randel and Wu (1999) reported that Antarctic ozone depletion cools the stratosphere and strengthens stratospheric westerlies and the polar vortex in early austral spring. This cooling extends to the troposphere, resulting in positive anomalies of the Southern Annular Mode (SAM) in the troposphere (Thompson and Wallace, 2000; Garfinkel et al., 2013). Consequently, during austral summer, this results in a poleward shift of the Southern Hemisphere subtropical jet stream (Son et al., 2010), shifts in rain and drought zones (Son et al., 2009; Kang et al., 2011; Polvani et al., 2011), a southward expansion of the Hadley cell (Garfinkel et al., 2015; Min and Son, 2013; Solomon and Polvani, 2016), and anomalies in ocean circulation (Sigmond and Fyfe, 2010; Bitz and Polvani, 2012; Seviour et al., 2019; Xia et al., 2020). In the Northern Hemisphere (NH), Arctic stratospheric ozone depletion also has significant impacts on the tropospheric climate and ocean systems (Cheung et al., 2014; Karpechko et al., 2014; Ma et al., 2019; Zhang et al., 2022). Specifically, the strengthened Arctic stratospheric polar vortex associated with ozone depletion increases temperature in the mid- and high-latitude regions of East Asia (Ivy et al., 2017) and the Hadley cells shifts toward the equator, leading to enhanced aridity in the subtropics (Hu et al., 2019a). Furthermore, stratospheric ozone depletion in the polar region can influence ocean systems. Ozone depletion leads to the strengthening of the Arctic polar vortex, triggering cyclonic surface circulation, leading to a reduction in Arctic sea ice in spring and summer (Zhang et al., 2022). Additionally, the enhanced Arctic polar vortex driven by ozone depletion results in negative anomalies in the North Pacific Oscillation (NPO) and negative sea surface temperature anomalies in the North Pacific and a positive Victoria mode, leading to El Niño-like sea surface temperature anomalies (Xie et al., 2016, 2017). These results suggest that the stratospheric polar vortex is a key circulation system in the interaction between stratospheric ozone and troposphere climate change.

Yet, the ozone-climate interactions remain not fully understood. Most studies have focused on one part of this feedback, such as the influence of the polar vortex on stratospheric ozone at interannual scales or in extreme years, especially in the Antarctic stratosphere, and there is less concern in the Arctic. Lin et al. (2021) pointed out that the dynamic response to ozone depletion drives the difference in the temperature response in Antarctica during austral spring and emphasized the importance of ozone-circulation coupling. In the NH, there has been a negative trend in stratospheric wave activity in late winter (Randel et al., 2002; Zhou et al., 2001), with no significant trends in early and midwinter (from November to January) from 1950–2000 (Hu and Tung, 2003). Additionally, the trends in stratospheric extratropical temperature show different signs between December compared to January and February from 1980–2000 (Bohlinger et al., 2014; Young et al., 2012). Previous literature mainly focused on the dynamic factors (e.g., planetary waves and Brewer-Dobson circulation, etc.) responsible for long-term temperature trends (Newman et al., 2001; Hu and Fu, 2009; Young et al., 2012; Ossó et al., 2015; Fu et al., 2019). However, whether the ozone-climate interactions can influence long-term temperature trends is still unclear. Therefore, in this study, we focus on the long-term trends in Arctic stratospheric temperature and what the contribution of the ozone-climate interaction is to these trends during winter and spring. Furthermore, what are the mechanisms responsible for the trends in different seasons? This study emphasizes the importance of ozone-climate interactions in the Arctic stratosphere, which can contribute to the development of future climate models. Section 2 outlines the data, methodologies,





and climate model experimental designs employed in this study. Section 3 presents the observed trends in temperature and ozone concentrations over the Arctic stratosphere, and Section 4 explores the underlying physical processes. Section 5
summarizes the conclusions and discusses future directions.

## 2 Data, methods and experimental configurations

### 2.1 Data

The European Centre for Medium-Range Weather Forecasts (ECMWF) v5 reanalysis dataset (ERA5; Hersbach et al., 2020) from 1980 to 2020 is used in this study. The horizontal resolution of this dataset is 1°×1° (latitude × longitude) and there are
37 vertical levels ranging from 1000 to 1 hPa. The daily and monthly mean results are derived from the 3-hourly ERA5 reanalysis dataset. We also used daily meteorological data obtained from the NASA Modern-Era Retrospective Analysis for Research and Applications version 2 (MERRA2) product (Gelaro et al., 2017), which has a horizontal resolution of 1.25°× 1.25° (latitude × longitude), and 42 pressure levels in the vertical direction extending from 1000 to 0.1 hPa from 1980 to 2020. The meteorological fields used in this study include daily mean horizontal winds, temperature, geopotential height and
ozone.

### 2.2 Methods

#### 2.2.1 Diagnosis of wave activity

##### 2.2.1.1 Elisassen-Palm flux

The Elisassen-Palm (E-P) flux (Andrews et al., 1987) is used to diagnose the propagation of waves in the vertical and
meridional directions and is calculated as follows:

$$F_\phi \equiv \rho_0 a cos\phi \left( \frac{\overline{u_z v'\theta'}}{\overline{\theta_z}} - \overline{u'v'} \right) \tag{1}$$

$$F_z \equiv \rho_0 cos\phi \left\{ \left[ f - (acos\phi)^{-1}(\overline{u}cos\phi)_\phi \right] \frac{\overline{v'\theta'}}{\overline{\theta_z}} - \overline{w'u'} \right\} \tag{2}$$

$$\nabla \cdot \vec{F} \equiv -\left( \rho_0 \overline{u'v'} \right)_\phi + \left( \rho_0 f \frac{\overline{v'\theta'}}{\overline{\theta_z}} \right)_z \tag{3}$$

where $\rho_0$ represents the density; $z$ represents the altitude; $a$ represents the radius of the Earth; $\phi$ represents the latitude; $f$
represents the Coriolis parameter; $\theta$ represents the potential temperature; $u$ and $v$ represent the zonal and meridional winds, respectively; and $w$ represents the vertical velocity. The overbars represent the zonal average, and the primes represent




deviations with respect to the zonal average. We ignore the term $\overline{w'u'}$ because it is small relative to the other terms (Zhang et al., 2019; Zhao et al., 2022).

### 2.2.1.2 Refractive index

The quasigeostrophic refractive index (RI) is used to diagnose the environment of wave propagation (Chen and Robinson, 1992) and is calculated as:

$$RI = \frac{\overline{q}_\varphi}{\overline{u}} - \left(\frac{k}{a\cos\varphi}\right)^2 - \left(\frac{f}{2NH}\right)^2 \tag{4}$$

where the meridional gradient of the zonal mean potential vorticity is calculated as:

$$\overline{q}_\varphi = \frac{2\Omega}{a}\cos\varphi - \frac{1}{a^2}[\frac{(\overline{u}\cos\varphi)_\varphi}{a\cos\varphi}]_\varphi - \frac{f^2}{\rho_0}(\rho_0\frac{\overline{u}_z}{N^2})_z \tag{5}$$

where $-\frac{f^2}{\rho_0}\left(\rho_0\frac{\overline{u}_z}{N^2}\right)_z = \left(\frac{f^2}{HN^2} + \frac{f^2}{N^4}\frac{dN^2}{dz}\right)\overline{u}_z - \frac{f^2}{N^2}\overline{u}_{zz}$ , and $H, q, k, N^2, \Omega, u_z$ are the scale height, potential vorticity, zonal wavenumber, buoyancy frequency, Earth's angular frequency, and zonal wind shear, respectively. The refractive index squared could be affected not only by the atmospheric zonal wind and wind shear but also by the quadratic vertical shear of the zonal mean zonal wind and atmospheric stability. As discussed in Matsuno (1970), it is expected that planetary waves of wavenumber $k$ tend to propagate toward regions where $n_k^2 > 0$ and are inhibited in regions where $n_k^2 < 0$.

### 2.2.1.3 The Brewer-Dobson circulation

The Brewer-Dobson circulation (BDC) driven by wave breaking in the stratosphere is closely related to stratospheric wave activity. The BDC in the atmosphere is represented in log-pressure coordinates as follows (Andrews et al., 1987):

$$\overline{v}^* \equiv \overline{v} - \rho_0^{-1}(\rho_0\overline{v'\theta'}/\overline{\theta}_z)_z \tag{6}$$

$$\overline{w}^* \equiv \overline{w} + (a\cos\phi)^{-1}(\cos\phi \cdot \overline{v'\theta'}/\overline{\theta}_z)_\phi \tag{7}$$

where $\overline{v}^*$ and $\overline{w}^*$ are the zonal-mean meridional and vertical velocities, respectively, $\theta$ is the potential temperature, $a$ is the radius of Earth, $\phi$ is the latitude, $\rho_0$ is the air density, and $z$ is the log-pressure height.

Using the generalized downward control principle, the BDC can be further decomposed into different forcing terms (Song and Chun, 2016):




$$\overline{v}^* = -\frac{1}{\rho_0 \cos\varphi}\frac{\partial}{\partial z}\left\{-\cos\varphi\int_z^\infty \rho_0\left[\frac{\frac{1}{\rho_0 a\cos\varphi}\nabla\cdot\mathbf{F}+\overline{\mathrm{GWD}}+\overline{X}-\frac{\partial\overline{u}}{\partial t}}{f-\frac{1}{a\cos\varphi}\frac{\partial}{\partial\varphi}(\overline{u}\cos\varphi)}\right]dz'\right\}$$ (8)

$$\overline{w}^* = \frac{1}{\rho_0 a\cos\varphi}\frac{\partial}{\partial\varphi}\left\{-\cos\varphi\left[\int_z^\infty \rho_0\left[\frac{\frac{1}{\rho_0 a\cos\varphi}\nabla\cdot\mathbf{F}+\overline{\mathrm{GWD}}+\overline{X}-\frac{\partial\overline{u}}{\partial t}}{f-\frac{1}{a\cos\varphi}\frac{\partial}{\partial\varphi}(\overline{u}\cos\varphi)}\right]dz'\right\}$$ (9)

where $\nabla\cdot\mathbf{F}, \overline{\mathrm{GWD}}, \overline{X},$ and $\partial\overline{u}/\partial t$ represent the E-P flux divergence, gravity wave forcing, residual term of the transformed Eulerian mean (TEM) equations, and zonal-mean zonal wind tendency, respectively. Song and Chun (2016) reported that the gravity wave drag term $\overline{\mathrm{GWD}}$ and the residual term $\overline{X}$ are relatively smaller than the E-P flux divergence

and zonal mean zonal wind tendency terms. Therefore, $\overline{\mathrm{GWD}}$ and $\overline{X}$ are not considered in this study.

**2.2.1.4 Takaya-Nakamura wave-activity flux**

The Takaya-Nakamura (T-N) wave activity flux (Takaya and Nakamura 1997; 2001; Nakamura et al., 2010) are used to represent the three-dimensional energy dispersion characteristics of the quasistationary Rossby wave with respect to climatological mean flow:


$$\mathrm{W} = \frac{p\cos\phi}{2|U|}\cdot\left\{\begin{array}{l}\frac{U}{a^2\cos^2\phi}\left[\left(\frac{\partial\psi'}{\partial\lambda}\right)^2-\psi'\frac{\partial^2\psi'}{\partial\lambda^2}\right]+\frac{V}{a^2\cos\phi}\left[\frac{\partial\psi'}{\partial\lambda}\frac{\partial\psi'}{\partial\phi}-\psi'\frac{\partial^2\psi'}{\partial\lambda\partial\phi}\right]\\[3mm]\frac{U}{a^2\cos\phi}\left[\frac{\partial\psi'}{\partial\lambda}\frac{\partial\psi'}{\partial\phi}-\psi'\frac{\partial^2\psi'}{\partial\lambda\partial\phi}\right]+\frac{V}{a^2}\left[\left(\frac{\partial\psi'}{\partial\phi}\right)^2-\psi'\frac{\partial^2\psi'}{\partial\phi^2}\right]\\[3mm]\frac{f_0^2}{N^2}\left\{\frac{U}{a\cos\phi}\left[\frac{\partial\psi'}{\partial\lambda}\frac{\partial\psi'}{\partial z}-\psi'\frac{\partial^2\psi'}{\partial\lambda\partial z}\right]+\frac{V}{a}\left[\frac{\partial\psi'}{\partial\phi}\frac{\partial\psi'}{\partial z}-\psi'\frac{\partial^2\psi'}{\partial\phi\partial z}\right]\right\}\end{array}\right\}$$ (10)

where the superscript is the zonal deviation and where $\phi, \lambda, \Phi, f = 2\Omega\sin\phi, a, \Omega$ are the latitude, longitude, geopotential height, Coriolis parameter, earth radius and Earth rotation rate, respectively. $\psi' = \frac{\phi'}{f}$ represents the perturbation of the quasiground transfer function relative to the climate field, and $U = (U, V)$ represents the climatological basic flow fields.




In Eqs. (1)–(10), the overbar denotes zonal-mean quantities, and the prime indicates departure from the zonal mean. The subscripts denote partial derivatives. The Fourier decomposition is used to obtain components $u',v', and\ \theta'$ in Eqs. (1)–(3) and components $\nabla\bullet F\ and\ \psi'$ in (8)–(10) with different zonal wave numbers.

### 2.2.2 Statistical methods

The trend is measured by the slope of a linear regression based on least squares estimation. We use a two-tailed Student's $t$ test to calculate the significance of the trend or perform a mean difference analysis. This paper measures the results of the significance test with $p$ values or confidence intervals. $p \leq 0.1$, $p \leq 0.05$, and $p \leq 0.01$ indicate that the trend or mean difference is significant at/above the 90%, 95%, and 99% confidence levels, respectively.

### 2.3 Model and experimental configurations

The F_1955-2005_WACCM_CN (F55WCN) component in the Community Earth System Model (CESM) Version 1.2.2 is used to verify the ozone-climate interaction in the Arctic stratosphere, with transient chemistry forcing fields (such as ozone, greenhouse gases (GHGs), aerosols). The F55WCN includes an active atmosphere and land, a data ocean (run as a prescribed component by simply reading sea surface temperature forcing data instead of running an ocean model) and sea ice. The model resolution is 1.9° latitude by 2.5° longitude, the model has 66 vertical levels and the model top level is 5.96 ×
$10^{-6}$ hPa. The chemistry module in F55WCN calculates the concentrations of different species and includes both gas phase and heterogeneous chemistry. The physics schemes in the F55WCN are based on those in the Community Atmosphere Model, Version 4 (CAM4; Neale et al., 2013).

To understand the causality of the ozone-circulation coupling, we perform model experiments to isolate the impact of ozone
changes on stratospheric dynamics and circulation. Two simulations (i.e., the control experiment and O3clm experiment) use identical boundary conditions and initial conditions. Both of the two experiments run from 1970–2020, and the first 10 years constitute the spin-up time. The control experiment uses fully interactive ozone chemistry, and long-term stratospheric ozone changes are involved in the radiation scheme. In contrast, in the O3clm experiment, the climatological mean ozone from the control simulation is imported into the radiation scheme, producing fixed radiative feedback, i.e., disabling the ozone-climate
interactions over a long period. This setting is designed to preserve the seasonal temperature changes caused by the real ozone radiation process, ensure stable operation of the experiment, and avoid introducing systematic biases. Thus, the comparison between the control and O3clm experiments can separate the feedback effects of long-term stratospheric ozone changes on atmospheric temperature and circulation (Fig. 1).



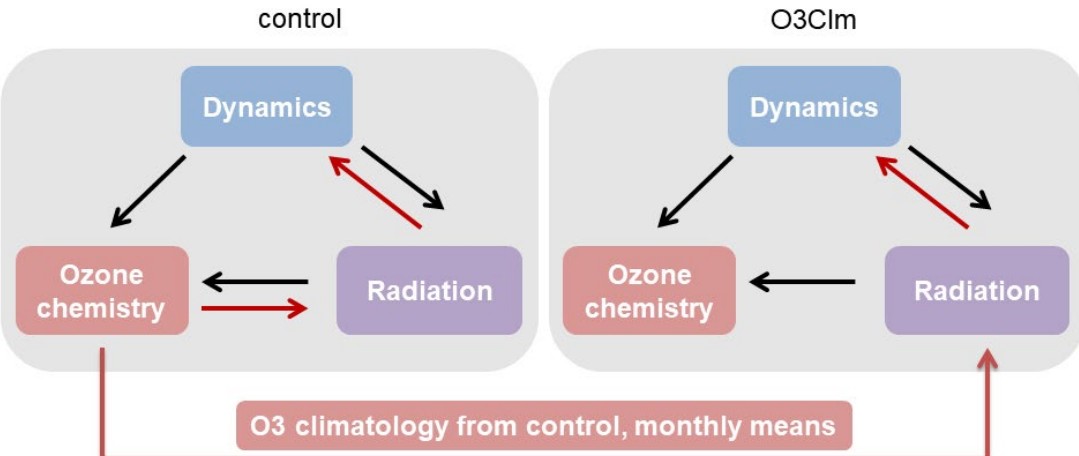

**Figure 1** Simulation setup of the control and O3clm experiments. The control experiment treats ozone chemistry fully interactively. That is, the calculated ozone field has a direct feedback on the atmosphere via the model radiation schemes. In contrast, the O3clm experiments do not use interactively calculated ozone in the radiation module. Instead, the radiation module uses an ozone climatology, which is derived from control runs with interactive ozone of the same model.

## 3 Trends in temperature and ozone over the Arctic in the middle and lower stratosphere

Figure 2 displays the normalized time series and linear trends in temperature in the Arctic stratosphere, averaged between 10 and 150 hPa, during different periods from late autumn to early spring. In November–December, the Arctic stratospheric temperature exhibit a small positive trend from 1980–1999 in both the MERRA2 and ERA5 datasets, and it shows an insignificant negative trend from 2000–2019 (Fig. 2a, the black line represents MERRA2; the gray line represents ERA5). This suggests that there is a warming trend in the Arctic stratosphere during early winter from 1980 to 1999, followed by a cooling trend from 2000 to 2019. The control experiment reproduces these trends well, with a significant positive trend in temperature from 1980 to 1999 and a significant negative trend from 2000 to 2019 (Fig. 2a, purple line). From January– February, the temperature displays an insignificant negative trend in both periods derived from the three datasets (Fig. 2b), suggesting that there is persistent cooling over the four decades. In March–April, the temperature shows a significant negative trend from 1980–1999 and no significant trend from 2000–2019 among the three datasets (Fig. 2c). Overall, the long-term changes in temperature derived from the control run align with the results from the reanalysis datasets. Comparing the trends from November–December and January–February, it is worth noting that there are contrasting temperature trends from 1980–1999, which change from positive to negative trends, whereas the temperature trends from 2000–2019 change from slight decreases to stronger declines. The reasons responsible for the intraseasonal opposite temperature trends are investigated in the following section. In this study, we primarily focus on a detailed analysis of the period from 1980–1999, during which significant stratospheric ozone depletion occurred (LOTUS, 2019; IPCC, AR6, 2023). The changes from 2000–2019 are discussed only in the final section.



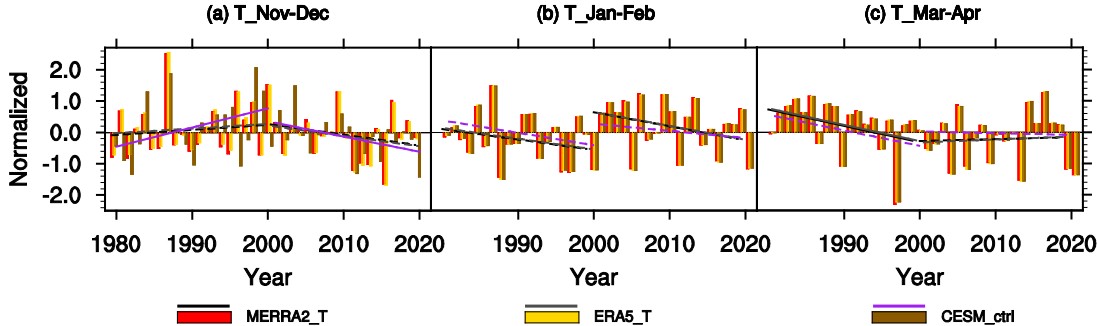

**Figure 2** Normalized time series of the temperature averaged from 150 hPa to 10 hPa over 65°–90°N from 1980–2020 in (a) November–
December, (b) January–February, and (c) March–April derived from MERRA2 (red column), ERA5 (orange column) and CESM (brown
column) control simulations. The color straight lines represent the linear trends before 2000 and after 2000. Solid lines indicate that the
trends are statistically significant at the 90% confidence level according to Student's *t* test

Figure 3 shows the trends in daily ozone and temperature between 10 and 250 hPa in the polar cap regions (65°–90°N) from
1980 to 1999, which are based on data from MERRA2 and the control run. The trend reversal is also evident in January,
which is consistent with Fig. 1. Before mid-January, there is an increasing trend in both ozone and temperature across all
levels (Fig. 3a, b). While after mid-January, the trends in temperature and ozone reverse in the middle and lower stratosphere.
Similar trend patterns are observed in the control run (Fig. 3c, d), indicating that the control run can reproduce the long-term
trends in stratospheric temperature and ozone in both early and late winter in the stratosphere. Therefore, it is reliable to use
the CESM model to analyze these trends in the following text.

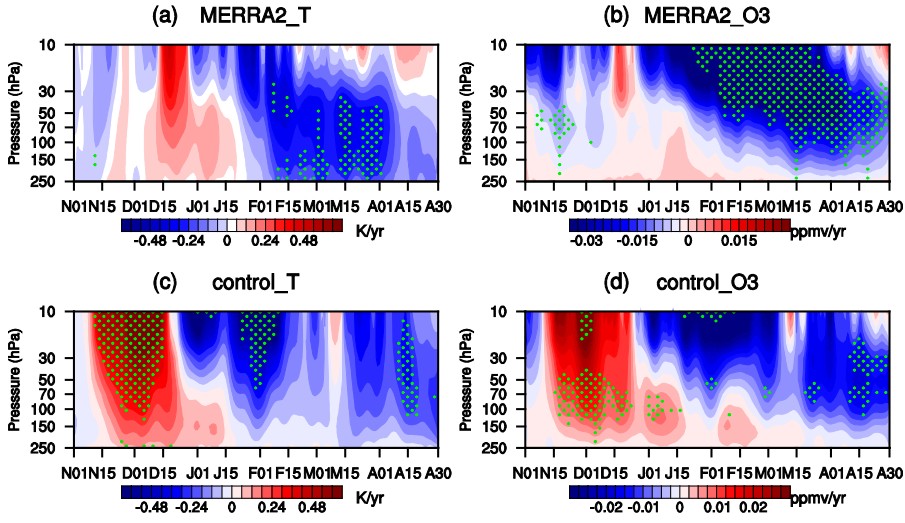



**Figure 3** Time evolution of trends in daily temperature (a, c) and ozone (b, d) in the polar cap regions (65°–90°N) during winter and spring derived from MERRA2 and the control run for the period of 1980–1999. The green dotted regions indicate that the trends are statistically significant at the 90% confidence level according to Student's *t* test.

Figure 4a and b display the daily trends in ozone and temperature between 10 and 250 hPa in the polar cap regions (65°–90°N) from 1980–1999 derived from the O3clm run (for the simulation set-up, see Methods). The O3clm run shows a nonsignificant temperature trend from November–December and a positive trend in late February and March. This result is completely opposite to that in the control run (Fig. 3b, c). The stratospheric ozone exhibits negative trends over the 10–100 hPa range from November to April, without an intraseasonal reverse. Between 100 and 150 hPa, there is a negative trend in early winter and a positive trend in early spring in the O3clm simulation. These trends are also opposite to those in the

control run (Fig. 3d). The differences in trends between the control run and O3clm run are shown in Fig. 4c, d. Note that there are significant positive anomalies in temperature and ozone trends from November–December and significant negative differences from late January, which are due to net ozone chemical-radiative-dynamical feedback effects. These differences suggest that ozone-climate interactions are crucial for long-term changes in stratospheric ozone and temperature.

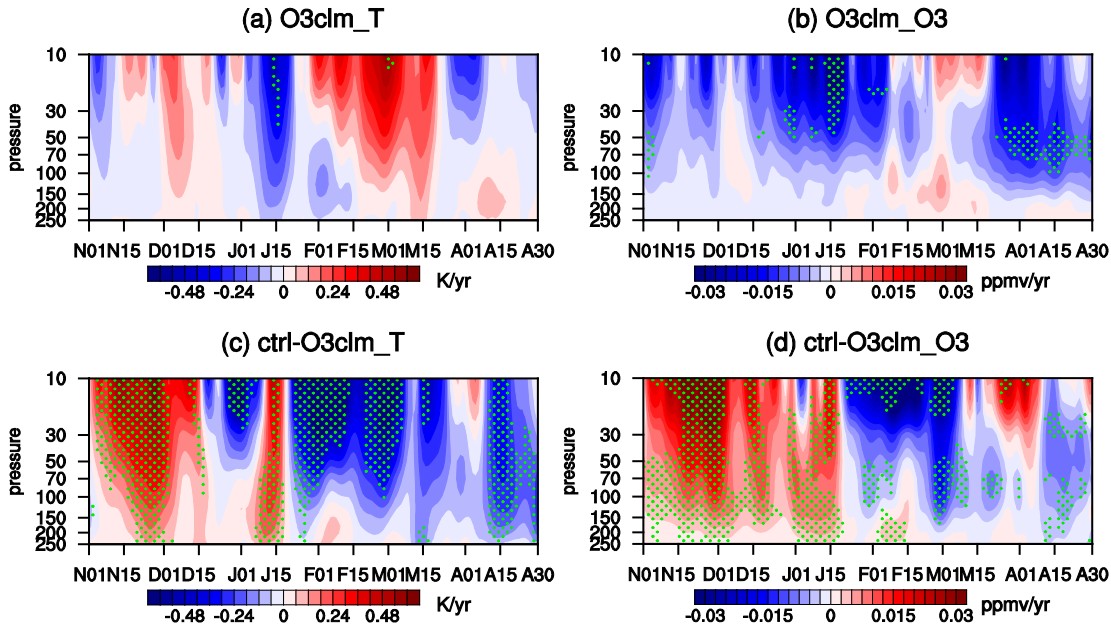


**Figure 4** Time evolution of the trend of daily temperature and ozone over the levels between 10 and 250 hPa in the polar cap (65°–90°N) during winter and spring derived from the O3clm run (a–b) and the differences between the control run and O3clm run (c–d) from 1980–1999. The green dotted regions indicate that the trend is statistically significant at the 90% confidence level according to Student's *t* test.



## 4 Trends in temperature and ozone over the Arctic in the middle and lower stratosphere

The core process of ozone-climate interactions is the ozone-circulation feedback. Figure 5 displays the trend in the vertical component downwelling branch of the BDC (w*) averaged over the subpolar region (50°–80°N) for the period of 1980–1999 in both the control run and O3clm run. We also decompose these trends into contributions from wave 1 (Fig. 5b, e and h) and wave 2 (Fig. 5c, f and i). The control run shows significant negative trends in w* from November to early December, corresponding to enhanced downwelling, and positive trends in w* from late December to January, corresponding to

weakened downwelling (Fig. 5a). After February, the trends in w* are less significant (Fig. 5a). The enhanced downwelling favors polar adiabatic warming, resulting in a positive temperature trend in late autumn and early winter, whereas in late winter, this situation reverses. Additionally, wave 1 dominates the trends in w*. In the O3clm run, there is no negative trend in w* in November and early December (Fig. 5d–f). This result indicates that ozone-circulation feedback strengthens the downwelling branch of the BDC, leading to adiabatic warming; conversely, there are anomalous upward motions that induce

anomalous adiabatic cooling from January to February, which is consistent with the reversal of the temperature trend in January (Figs. 3, 4). The differences between the control run and O3clm run suggest a similar pattern to that of the control run (Fig. 5g, h and i). Overall, the changes in the downwelling branch of the BDC during November and early December are mainly modulated by the ozone-climate interactions. The results suggest that adiabatic warming due to the strengthening of the downwelling branch of the BDC plays a crucial role in Arctic temperature from November to early December. Similar

results have been reported in previous studies (Albers and Nathan, 2013; Hu, et al., 2019b).

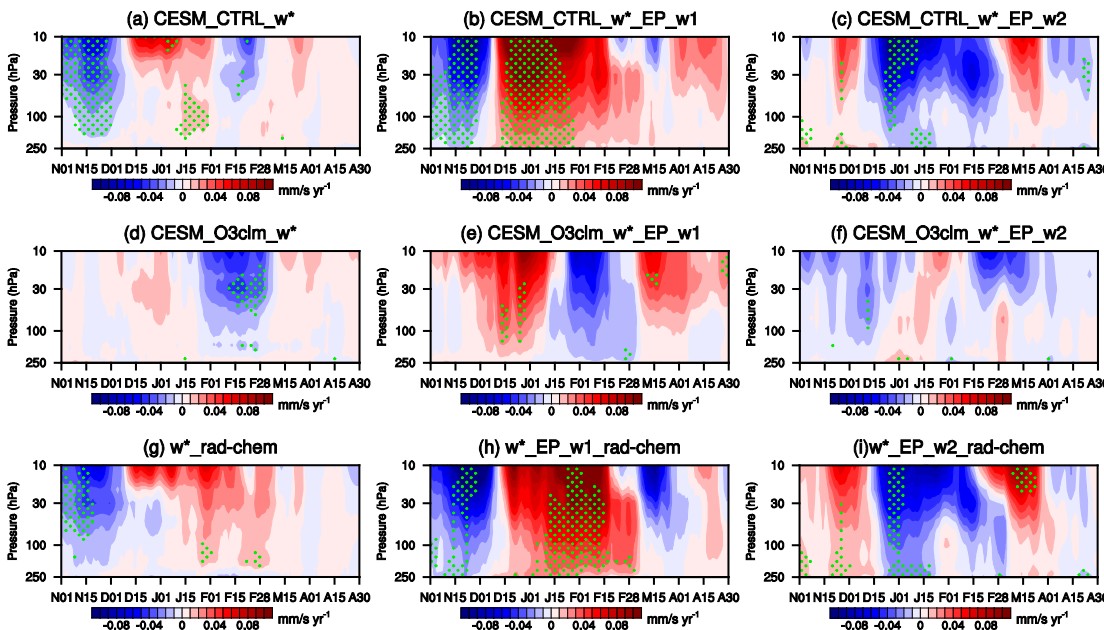

**Figure 5** Linear trend of the vertical component (w*) of the BDC and its contribution (shading) to the wavenumber 1 (b, e and h) and wavenumber 2 components (c, f and i) from 1980–1999 averaged in the subpolar region (50°–80°N) during winter and spring, derived



from the control run (a, b and c), O3clm run (d, e amd f) and the differences between the control run and O3clm run (g, h and i). The green
stippled regions indicate the trend of the BDC significant at/above the 90% confidence level according to Student's *t* test.

Anomalous BDC trends associated with ozone-climate interactions can be attributed to the upward planetary waves. Figure 6
shows the trends in stratospheric planetary wave activity over the subpolar region (50°–80°N) from November–April. In the
control run, there is a significantly positive trend in the waves entering the stratosphere in November and early December
from 1980–1999, which is accompanied by intensified wave flux convergence in the middle stratosphere (approximately 10–
50 hPa; Fig. 6a). However, in late December and January, the waves entering the stratosphere decrease, accompanied by
weakened wave flux divergence. These features imply that stratospheric planetary wave activity strengthened in November
and early December and weakened in late December and January during the 1980–1999 period, which is consistent with the
findings of previous studies (Bohlinger et al., 2014; Young et al., 2012). In contrast, in the O3clm run, waves entering the
275   stratosphere in November and early December decrease, and there is no significant convergence trend from 1980–1999 (Fig.
6d). The trends in the planetary wave are mainly contributed by the wave 1 component rather than by wave 2 (Fig. 6b, c, h
and i). In November and December, the enhanced planetary wave moves upward, and the convergence of the E-P flux
weakens the westerlies and increases the temperature in the lower stratosphere, which is consistent with the enhanced
downward motions shown in Fig. 5. The trends of planetary wave activity and E-P flux convergence in January and February
280   are opposite to those in early winter. Overall, the feedback of upward wave propagation and BDC plays a crucial role in
reversing the stratospheric temperature trend at the intraseasonal timescale during winter. Notably, the planetary wave
activity only changes noticeably before February in the control run and O3clm run, and then gradually weakens.

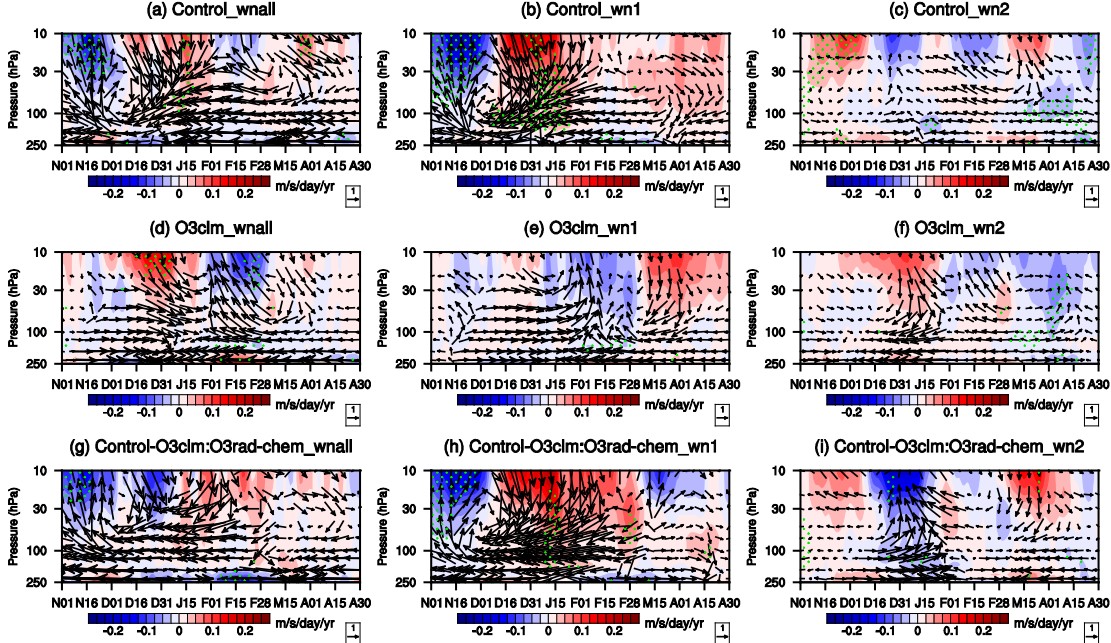



**Figure 6** Trends in E-P flux (a, d and g; arrows; units of horizontal and vertical components are $10^4$ and $10^2$ kg s$^{-2}$ yr$^{-1}$, respectively; an arrow pointing to the right indicates poleward propagation, whereas an arrow pointing to the left indicates equatorward propagation) and its divergence (shading) with their wave 1 components (b, e and h) and wave 2 components (c, f and i) over the levels between 10 and 250 hPa from 1980–1999 averaged in the subpolar region (50°–80°N) during winter and spring, as derived from the control run (a–c), O3clm run (d–f) and the differences between the control run and O3clm (g–i). The green stippled regions indicate the trend of the E-P flux divergence significant at/above the 90% confidence level according to Student's *t* test.

Previous studies emphasized that planetary waves entering the stratosphere are primarily modulated by propagating conditions in the upper troposphere and lower stratosphere regions (Albers and Nathan, 2013; Hu, et al., 2019b). The refractive index (RI) is a good metric for assessing the atmospheric state for planetary wave propagation. Theoretically, regions with a larger RI are more favorable for planetary wave propagation (Andrews et al., 1987). On the basis of the formula of the RI (see Equation (4) in the Methods section), the second term of the RI is a constant with specific wavenumbers, and the third term of the RI is negligible compared to the changes in the first term (Simpson et al., 2009; Hu, et al., 2019b; Hu, et al., 2022). It is suggested that variations in the meridional gradient of zonal mean potential vorticity ($\overline{q}_\varphi$) could account for most of the changes in the RI at mid- and high- latitudes (Simpson et al., 2009; Zhang et al., 2020). Figure 7 shows the daily evolution of the trend in the RI, the vertical component of the E-P flux and $\overline{q}_\varphi$ in the middle and lower stratosphere (30–150 hPa) averaged between 45° and 75°N from November to February from 1980–1999. The datasets are derived from the control run and O3clm run. In the control run, significant positive trends in the RI persist until mid-December in the middle and lower stratosphere (black lines), implying that more planetary waves could enter the stratosphere due to ozone-climate interactions. This corresponds to the strengthened vertical component of the E-P flux in the stratosphere (purple line; and Fig. 6a). Higher $\overline{q}_\varphi$ values (blue line) lead to a larger RI, providing favorable atmospheric conditions for upward wave propagation. However, after mid-December, the RI trends become negative in the middle and lower stratosphere, suppressing upward wave propagation, which is consistent with the reduced E-P flux during this period (Fig. 6a). There is a remarkable reversal of $\overline{q}_\varphi$ in late December. The negative $\overline{q}_\varphi$ trend persists until February in the middle and lower stratosphere, which basically corresponds to a negative trend in the RI, which consequently affects the intraseasonal reversal signal in the E-P flux (Fig. 6b). Therefore, changes in $\overline{q}_\varphi$ could serve as the main factors influencing the changes in the RI, consequently impacting the propagation of planetary waves. In the O3clm run, the RI reveals slight negative trends in early November. This finding indicates that planetary waves are more likely to be reflected in the middle and lower stratosphere. However, from late November to mid-January, the RI exhibits positive trends, and after mid-December, the RI trends become negative until mid-January. The results derived from the O3clm run are markedly different and almost opposite those derived from the control run. Overall, the results indicate the impact of ozone-climate interactions resulting from ozone depletion on wave propagation conditions. The feedback of ozone-climate interactions triggered by



ozone depletion could modulate stratospheric temperature and zonal winds, influencing $\overline{q}_{\varphi}$ and the RI, which play a key role in upward wave propagation during early winter.

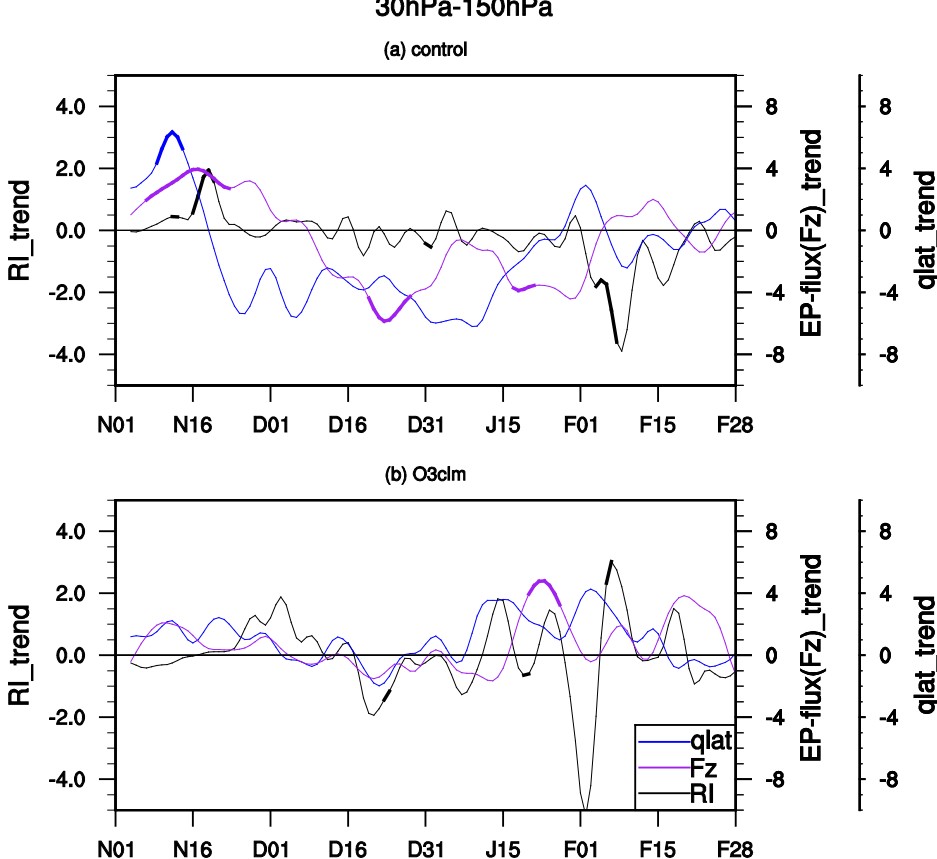

**Figure 7** Daily evolution of the trends in the RI, vertical component of the E-P flux and $\overline{q}_{\varphi}$ from 1980-1999 at 50–150 hPa from 1 November to 28 February, derived from the control run (a) and O3clm run (b). The solid lines indicate the trends in the significant RI, vertical component of the E-P flux and $\overline{q}_{\varphi}$ at/above the 90% confidence level according to Student's $t$ test.

Figure 8 shows the trends in the zonal wavenumber 1 geopotential height and climatology from November to January, as well as the difference in the T-N wave activity flux between the low-ozone (1980–1985) and high-ozone (1997–2002) periods. In the control run, the climatological mean displays a westward-tilted structure with increasing height. In November, the zonal wavenumber 1 geopotential height tendency is somewhat in phase with the climatological mean, indicating an amplification of wavenumber 1 baroclinic waves, particularly in the lower stratosphere. Furthermore, the T-N wave flux propagates upward to the eastern hemisphere, causing the wave center of the geopotential height to move eastward. In





December, the positive center of the geopotential height anomalies shifts from 90°W to 0°, and the negative center shifts
from 90°E to 180°, indicating that the geopotential height anomalies have shifted eastward by approximately 90° compared
with those in November. As a result, the geopotential height anomalies are out of phase with the climatological mean,
corresponding to weakened upward wave propagation in December. Note that there remains enhanced eastward wave
propagation in the eastern hemisphere in the upper stratosphere during this month, leading to a further 60° eastward shift in
geopotential height anomalies and a continuous weakening of stratospheric planetary wavenumber 1 in January.
Correspondingly, an intra-seasonal reversal signal in the E-P flux is observed in the control run (Fig. 6).

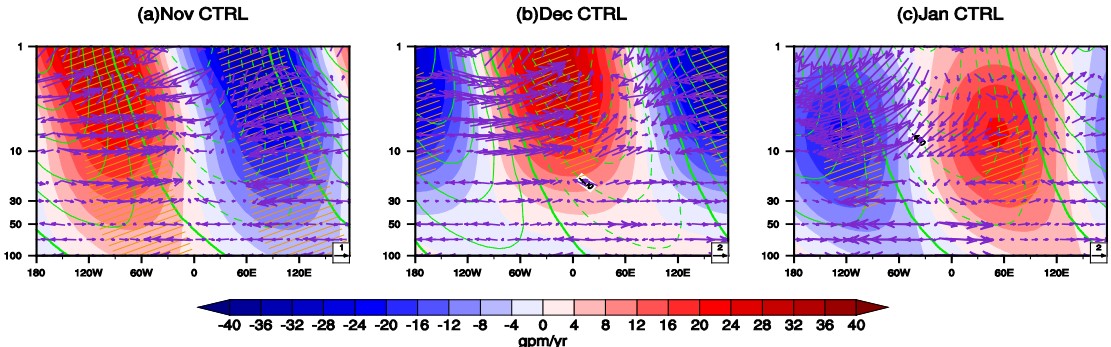

**Figure 8** Height-longitude cross sections along 60°N of the zonal wave-1 geopotential height trend from 1980-1999 (shaded areas) and the
difference in T-N flux between the low-ozone period (1980-1985) and high-ozone period (1997-2002) (vector arrows). The green contours
indicate the climatological mean of the zonal wave-1 geopotential height derived from the control run. The orange lines represent values
that are statistically significant at the 90% confidence level according to Student's *t* test.

In addition, the propagation of planetary wavenumber 1 can drive the Arctic stratospheric polar vortex toward Eurasia and
promote polar vortex shift events (Mitchell et al., 2011; Zhang et al., 2016; Huang et al., 2018). The impacts of ozone-
climate interactions on the position of the Arctic stratospheric polar vortex remain unclear. Figure 9 depicts the potential
vorticity (PV) differences and stratospheric polar vortex edge between the low- and high-ozone periods, derived from the
control and O3clm runs. In the control run, there are positive PV anomalies in eastern Eurasia during November and
December in the low-ozone period, along with a shift in the position of the polar vortex edge toward eastern Eurasia. In
contrast, in the O3clm run, the polar vortex edge remains more stable during the low-ozone period than during the high-
ozone period, and there are no significant changes in the PV anomaly in eastern Eurasia. These findings further indicate that
the ozone-climate interactions play a significant role in amplifying planetary wavenumber 1 (as shown in Figs. 6, 8), thereby
influencing the shift in the polar vortex during early winter. The results indicate that the dynamic feedback of ozone-climate
interactions significant influences on the position of the polar vortex in early winter. Figure 9e, f shows the temperature
differences averaged in November and December between the low-ozone and high-ozone periods, which are derived from
the control and O3clm runs. In the control run, there is a dipolar structure in the temperature anomalies in the middle and
lower stratosphere (30–100 hPa). The temperature experiences a significant decline over eastern Eurasia in the control run





compared with that in the O3clm run, accompanied by a stronger decline in this region than in other regions at the same latitudes, corresponding to the shift in the polar vortex (Fig. 9a, c). In contrast, in the O3clm run, there is a single structure in the temperature anomalies. In addition, the results of the polar vortex shift are consistent with the enhanced stratospheric planetary wave activity, which reconfirms that the ozone-climate interactions can modulate stratospheric temperature

through dynamic processes in early winter, which is consistent with the findings of Zhang et al. (2020).

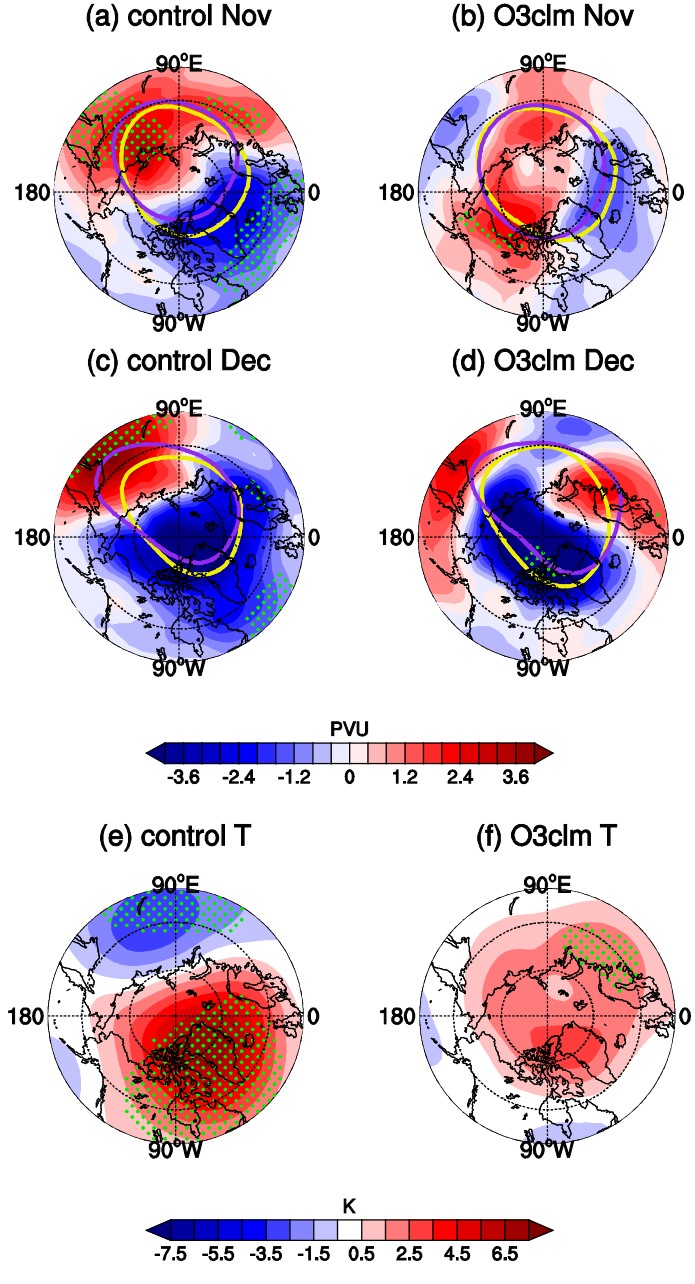





**Figure 9** Differences in PV between the low-ozone period (1997–2002) and high-ozone period (1980–1985) averaged between 430 and 600 K (shaded) in November (a, b) and December (c, d) derived from the control run (a, c) and O3clm run (b, d). The edges of the polar vortex in the low-ozone period and high-ozone period are represented by purple lines and yellow lines, respectively. The differences in temperature between the low-ozone and high-ozone periods averaged between 30 and 100 hPa (shaded), averaged between November and December, which are derived from the control (e) and O3clm (f) runs. The green dotted regions denote that the difference of PV is statistically significant at the 90% confidence level according to Student's *t* test.

It is then natural to ask why there are distinct temperature responses in Arctic winter, in the absence of solar radiation, between the two experiments with and without ozone-climate interactions. Figure 10 shows the evolution of shortwave heating rate (referred to as the QRS) and longwave heating rate (referred to as the QRL) from November to April in the two experiments. Both the control and O3clm runs reveal relatively weak QRS trends from November to mid-February because sunlight cannot reach the Arctic region. In the control run, the QRL heating from November to early December shows a negative trend corresponding to the longwave cooling effect. In contrast, in the O3clm run, the ozone-climate interactions are removed and there is no significant QRL trend. The QRL cooling in the control run occurs because a warmer air parcel corresponding to the positive temperature trend in early winter emits more longwave radiation and hence cools faster. Lin and Ming (2021) noted that radiaitive damping due to longwave cooling could induce ozone-circulation interactions by increasing wave dissipation and modulating stratospheric circulation. Unlike their work, which focused on the ozone-circulation feedback processes in the Antarctic stratosphere, the present study offers more details on these processes in the Arctic winter stratosphere.

After February, the upward propagation of planetary waves and ozone-circulation feedback processes weaken, whereas the contribution of shortwave radiative processes to stratospheric temperature increases as sunlight reaches the Arctic region. The control run demonstrates that the ozone QRS shows a significant negative trend during the ozone-depletion period, which leads to a lower temperature and an intensified polar vortex (Brasseur and Solomon, 2005). However, in the O3clm run, the radiative effects of ozone-climate interactions are inactivated, leading to insignificant changes in QRS throughout the entire winter and spring. In addition, negative temperature anomalies (Fig. 2c and Figs. 3a, c) correspond to the colder air parcel emitting less longwave radiation and causing warming to generate positive QRL anomalies in spring. In the differences between the control run and O3clm run, QRS and QRL exhibit the same pattern as in the control run. Our results demonstrate that the ozone-climate interactions during early winter mainly influence stratospheric temperature through dynamic adjustments. In contrast, the trends in temperature during late winter and spring are primarily due to dynamic cooling and shortwave cooling overwhelming the longwave heating of radiation processes.



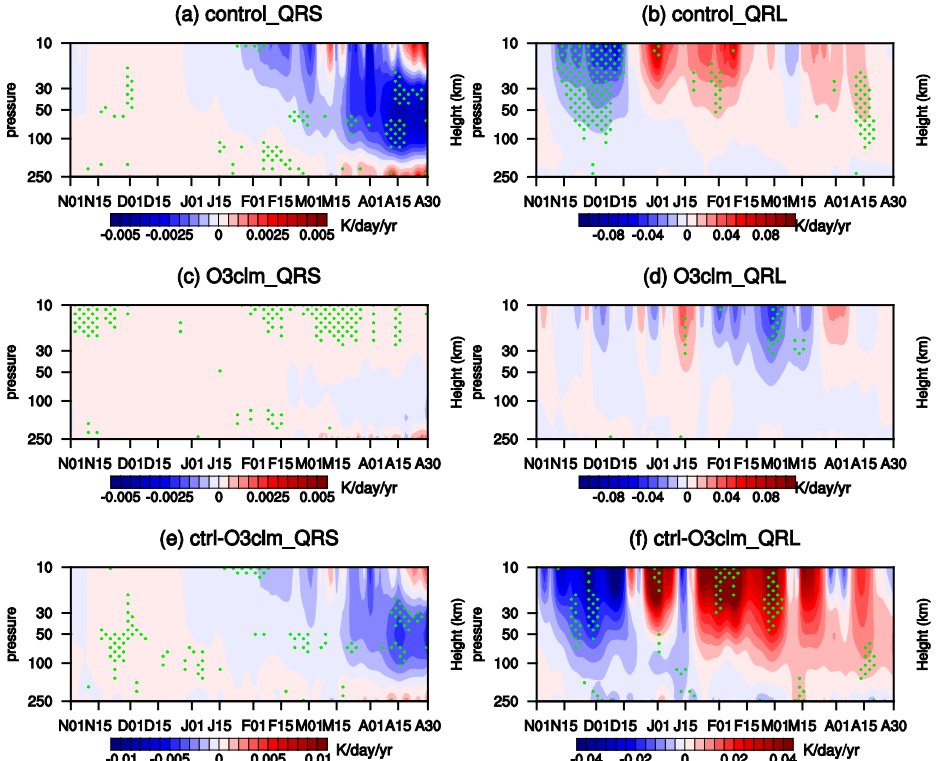

**Figure 10** Time evolution of trends in the daily shortwave heating rate (solar heating rate; QRS) and longwave heating rate (QRL)
between 10 and 250 hPa in the polar region (65°–90°N) during winter and spring derived from the control run (a, b), O3clm run (c, d) and
the differences between the two runs (e, f) from 1980–1999. The green dotted regions indicate that the trends are statistically significant at
the 90% confidence level according to Student's *t* test.

In the previous sections, we revealed the impact of ozone-climate interactions on stratospheric temperature and circulation
during the ozone-depletion period before 2000. To understand how ozone-climate interactions work after 2000, Figure 11
further illustrates the trend in the daily variation in ozone and temperature between 10 and 250 hPa in the polar cap regions
(65°–90°N) from 2000 to 2019, on the basis of MERRA2 data. The results show an unremarkable decrease in ozone and
temperature trends between 10 and 150 hPa during November. However, in December, there is a significant increasing trend
in ozone across all levels and a slightly positive trend in temperature (Fig. 11a, b). After February, the temperature and ozone
in the regions of the middle and lower stratosphere show significant negative trends. These changes are similar to those
before 2000, with the difference being that the reversal of the negative trend occurs earlier, in late December. Compared with
the pre-2000 period, there are positive anomalies for ozone and temperature in the middle and upper stratosphere in April
after 2000, indicating that the pre-2000 period experienced greater stratospheric ozone depletion (WMO, 2022).



Figure 12 shows the results derived from the control run, O3clm run, and the difference between the two runs. The control run shows an insignificant positive temperature trend at 100–250 hPa, and a nonsignificant negative anomaly in temperature in November and December, which is similar to the MERRA-2 results. In the O3clm run, all the original negative trends in the lower and middle stratosphere turn into positive trends. Furthermore, the differences in temperature and ozone between the control run and O3clm run look somewhat like the pre-2000 results, except for the early trend reversal in December, and

the differences are not significant most of the time. This suggests that the ozone-climate interactions continue to work after 2000, leading to intra-seasonal reversal trends in stratospheric ozone and temperature. Furthermore, in the post-2000 period, the negative ozone trends in March and April may delay ozone recovery during spring through the shortwave radiative cooling effect.

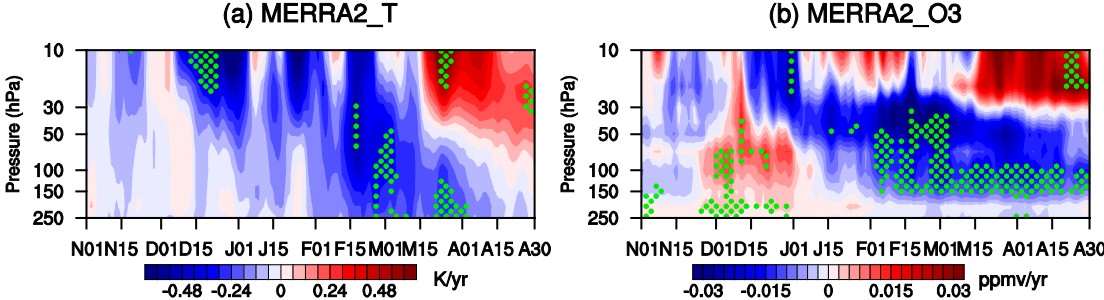

**Figure 11** Time evolution of trends in daily temperature (a) and ozone (b) over the levels between 10 and 250 hPa in the polar cap regions (65°–90°N) during winter and spring derived from MERRA2 from 2000–2019. The green dotted regions denote that the trends are statistically significant at the 90% confidence level according to Student's *t* test.

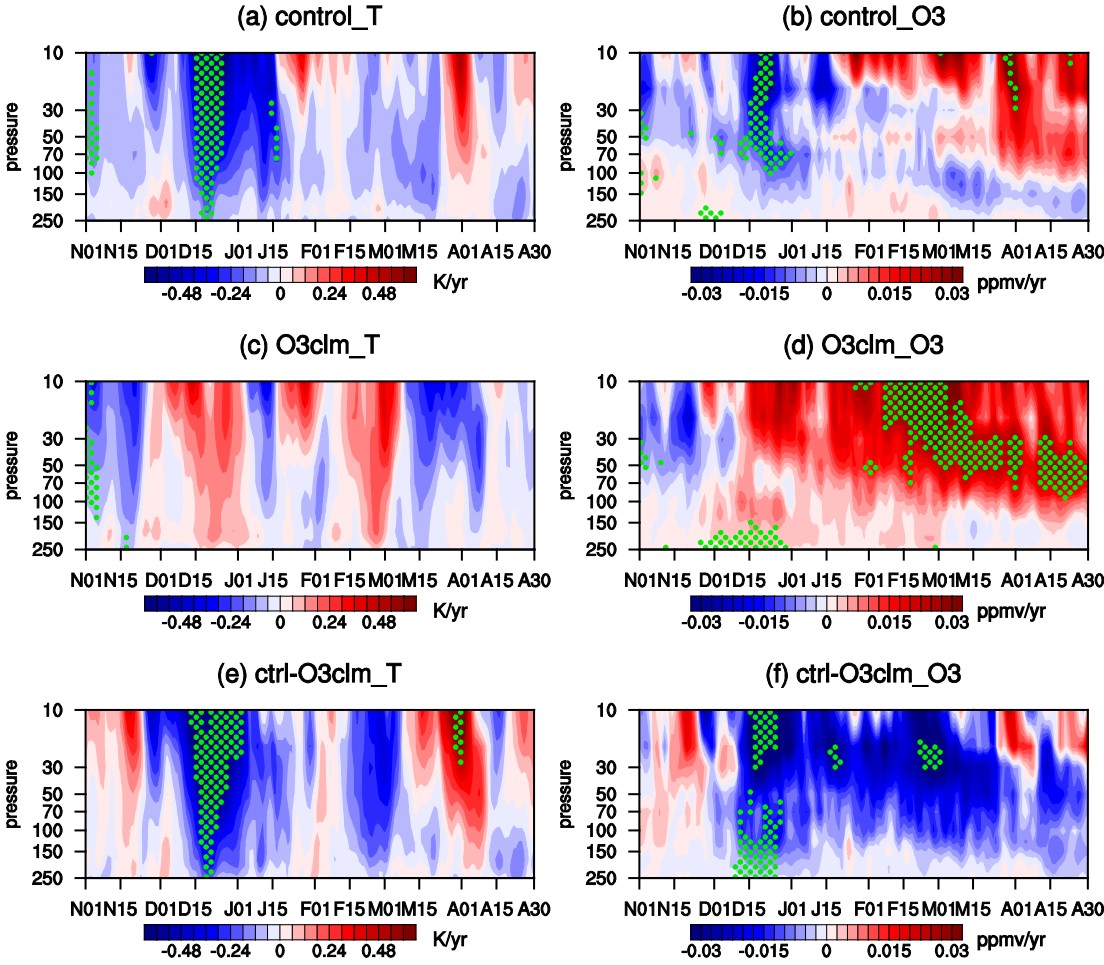

**Figure 12** Time evolution of the trends in daily temperature (a) and ozone (b) over the levels between 10 and 250 hPa in the polar cap
regions (65°–90°N) during winter and spring derived from the differences between the control run and O3clm run from 2000–2019. The
green dotted regions indicate that the trends are statistically significant at the 90% confidence level according to Student's *t* test.

## 5 Conclusion and discussion

This study investigates the impacts of ozone-climate interactions on the temperature trends in the Arctic stratosphere during
winter and early spring, using reanalysis datasets and CESM model simulations. We found that stratospheric Arctic
temperature in early winter (November and December) significant increases from 1980–1999 period (Figs. 2, 3 and 4),
which is primarily driven by enhanced planetary wave propagation into the stratosphere and a strengthened BDC. Notably,



the ozone-circulation feedback of ozone-climate interactions plays a key role in modulating this trend. Specifically, in early winter, ozone-circulation feedback can create an atmospheric state favorable for upward wave propagation and E-P flux convergence (Figs. 6, 7), which can also lead to a strengthened BDC (Fig. 5) and thereby a positive trend in temperature during early winter. These trends in the BDC and planetary wave activity are predominantly driven by planetary wavenumber 1 (Figs. 5, 6). In addition, we found that the spatial pattern of the temperature anomalies in the middle and lower stratosphere shows a zonally asymmetric structure and we further explore the reasons for this structure. The ozone-circulation feedback causes the T-N wave activity to propagate eastward and upward, resulting in the enhancement and eastward shift of the planetary wavenumber 1 (Fig. 8). This feedback causes the polar vortex to shift toward the eastern Eurasia continent and leads to a lower temperature over this region than over the other regions at the same latitudes (Fig. 9). During late winter and spring, there are negative trends in stratospheric Arctic temperature. Especially in early spring, when solar radiation reaches the pole region, ozone shortwave cooling during the ozone-depletion period plays a crucial role in these negative trends (Fig. 10). After 2000, the stratospheric temperature response to ozone changes is weaker than that from 1980 to 1999 (Figs. 11 and 12).

The ozone-climate interactions are critical processes in modulating these trends. Similar to earlier findings, our study highlights the role of planetary wave activity and BDC in influencing Arctic stratospheric temperature. However, our study highlights the dynamic feedback mechanisms driven by ozone-climate interactions, providing a new perspective on temperature circulation feedback, which has not been extensively explored in previous studies. Notably, various factors may influence the ozone-climate interactions. These factors include changes in greenhouse gas concentrations, nitrous oxide, volcanic activity, or other atmospheric constituents that influence radiative and chemical processes in the stratosphere (Eric Klobas et al., 2017; Meul et al., 2016; Ravishankara et al., 2009; Revell et al., 2015; Solomon et al., 2009). This raises questions about other potential feedback mechanisms for ozone-climate interactions in the future. Future studies are needed to better understand how and to what extent these factors can influence the ozone-climate interactions. Additional, it is essential to acknowledge several limitations in our study. Methodologically, reliance on model simulations introduces inherent uncertainties. For example, changes in experimental conditions may affect the robustness of our results. A more complete solution to these limitations may require us to conduct longer historical simulation experiments in the future to reduce experimental uncertainties. This study contributes to a better understanding of the role of ozone in Arctic stratospheric temperature dynamics, offering valuable insights for the development of climate models. Improved models could enhance predictions of stratospheric temperature changes, informing strategies for ozone protection and climate change mitigation.



**Acknowledgments**

This work is supported by the National Natural Science Foundation of China (42075062, 42130601). We also thank the scientific team at National Center for Atmospheric Research (NCAR) for providing the CESM-1 model. Finally, we thank
the computing support provided by Supercomputing Center of Lanzhou University.

**Author contributions**

JZ provided ideas and formulation or evolution of overarching research goals and aims, SZ conducted experiments, produced figures, and organized and wrote the paper. JZ, CZ and ZW contributed to the revisions made to the paper. CZ helped to design the experiments.

**Competing interests**

The contact author has declared that neither they nor their co-authors have any competing interests.

**Date Availability Statement**

The European Centre for Medium-Range Weather Forecasts (ECMWF) version 5 reanalysis dataset (ERA5) are openly available at https://cds.climate.copernicus.eu/cdsapp#!/dataset/reanalysis-era5-pressure-levels?tab=overview. The MERRA2
data are obtained from https://disc.gsfc.nasa.gov/datasets/M2I3NPASM_5.12.4/summary?keywords=%22MERRA-2%22. The CESM model is available at https://www2.cesm.ucar.edu/models/current. The data generated in this work can be obtained by contacting Siyi Zhao (120220900830@lzu.edu.cn).

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
