# Peer review of "Effects of Ozone-Climate Interactions on the Long-Term Temperature Trend in the Arctic Stratosphere"

_EGUsphere, 2024_

## Referee Comment (RC1)

**Review: Effects of Ozone-Climate Interactions on the Temperature Variation**

**in the Arctic Stratosphere**

by Siyi Zhao, Jiankai Zhang, Chongyang Zhang, Zhe Wang

I appreciate the authors efforts to isolate the impact of past Arctic ozone changes on trends in temperature and dynamics and to underline their findings with mechanical explanations. Unlike other studies, this study focuses on the extended winter season and might therefore extent our knowledge on the impacts of ozone changes from springtime to the whole winter season. However, I have doubts around the entire experiment setup (especially the way the ozone climatology is calculated) and think that the conducted simulations are not suitable to achieve the goal of this study. Moreover, the authors make use of a figure from another study without mentioning it.

Main comment:

More details should be provided on the ozone climatology that is used in the O3clim experiments. Are these daily or monthly means? 3D or zonal mean? If I understand it correctly from Fig. 1, then the ozone chemistry is still calculated in the O3clim setup, but it is not radiatively active. This information is missing in the experiment description in lines 179 ff. How many ensemble members were simulated? Moreover, Fig. 1 is identical to Fig. 1 from Friedel et al. (2022b) and the caption accompanying this figure is almost identical to the caption of Fig. 1 in Friedel et al. (2022a). This must be cited!

If I understand the experiment design correctly, I believe that the conducted simulations are not suitable to study the impact of past ozone changes on trends in temperature and dynamics. The authors contrast one simulation with fully interactive ozone for the period 1980-2020 against a simulation where they impose an ozone climatology calculated over this period. Then, they contrast trends from 1980-2000 in the two simulations to isolate the effect of ozone changes during that period on the variables of interest. I believe that this experiment design does not allow to disentangle the effect of past ozone changes, as the climatology was calculated over a period that includes ozone trends. Therefore, the ozone climatology used in the O3clim run might show lower ozone concentrations (due to an ozone decline from 1980-2000) than the ozone field in the crtl run in 1980. Hence, at the starting point of the simulations (i.e. 1980), the circulation/temperature and ozone field are not consistent in the O3clim simulation. Therefore, temperature changes might arise to adjust to the lower ozone concentrations and trends in this simulation cannot solely be attributed to dynamical and thermodynamical processes other than ozone. Moreover, with the approach used here, effects of potential long-term ozone changes and interannual ozone changes are mixed, since they are both disabled in the

O3clim experiments. When only using one ensemble member of 20 years, the role of interannual ozone variability is likely not negligible.

In order to achieve the goal posed by this study, I suggest first deriving an ozone climatology using a time-slice simulation with fixed boundary conditions (i.e. CFCs, GHGs,...) of the year 1980 with fully interactive ozone. The ozone climatology calculated this way can then be used to simulate the O3clim experiments from 1980-2000. Using this approach, the authors would be able to attribute changes in dynamics and temperature over this period to changes in ozone. However, more than one ensemble member for the transient runs would be necessary in order to isolate the effect of long-term ozone changes rather than the effect of interactive ozone variability.

Other comments:

- The introduction does not follow a clear outline. Sometimes it is unclear whether the authors are referring to interannual or long-term processes. The introduction reviews mostly literature that focuses on ozone-climate coupling on interannual timescales, but then the study focuses on long-term trends. Please tailor the introduction a bit better to the aim of this study. Some of the statements made in the introduction are also not supported by the cited literature (see detailed comments below). The title of the study is also misleading, as is suggests that this study focuses on interannual variability rather than long-term trends.
- Line 8: "increased"
- Line 15: "cooling trends in the Arctic stratosphere are ..."
- Line 35: Please be a bit more precise here. Marsh et al. (2016) conclude that ozone feedbacks are not crucial for the model's climate sensitivity in 4xCO2 forcing experiments. Maybe it is worth citing Rieder et al. (2019) in this context, who showed that ozone variations are important for the stratospheric temperature variability in models.
- Lines 44 ff: I could not find any evidence for this statement (i.e. the longwave ozone effect on stratospheric dynamics during winter) in the cited study (Strahan et al., 2013). Could you please provide further literature on the mid-winter longwave radiative effect of ozone?
- Lines 47 ff: Are you now talking about mid- or late-winter/spring? I believe this statement is only valid for springtime, when the polar vortex is already weakened. See also Haase and Matthes (2019).
- Line 85 ff: I do not really understand this sentence (starting with "Lin et al. ... "). Could you please reformulate?
- Lines 91 ff: Chiodo et al. (2023) studied the impact of long-term ozone trends in the Arctic on temperature and dynamics. Might be worth mentioning this here.
- Lines 94 ff. "..., we focus on the historical long-term trends ..."

- Lines 202 ff: To support this statement on the persistent cooling, you would have to calculate a trend over the whole time period. When doing this, you would probably get no trend, e.g. no significant temperature changes from 1980-2020.
- Fig. 2: How have the time series been normalised?
- The trends in Fig. 2 look a bit constructed, i.e. they seem to be very sensitive to the time chosen time frame. Especially the trends in springtime (March/April) seem to be entirely caused by the ozone depletion event in 1997.

References:

Chiodo, G., Friedel, M., Seeber, S., Domeisen, D., Stenke, A., Sukhodolov, T., and Zilker, F.: The influence of future changes in springtime Arctic ozone on stratospheric and surface climate, Atmos. Chem. Phys., 23, 10451–10472, https://doi.org/10.5194/acp-23-10451-2023, 2023.

Friedel, M., Chiodo, G., Stenke, A. *et al*. Springtime arctic ozone depletion forces northern hemisphere climate anomalies. *Nat. Geosci*. **15**, 541–547, https://doi.org/10.1038/s41561-022-00974-7, 2022a.

Friedel, M., Chiodo, G., Stenke, A., Domeisen, D. I. V., and Peter, T.: Effects of Arctic ozone on the stratospheric spring onset and its surface impact, Atmos. Chem. Phys., 22, 13997–14017, https://doi.org/10.5194/acp-22-13997-2022, 2022b.

Haase, S. and Matthes, K.: The importance of interactive chemistry for stratosphere–troposphere coupling, Atmos. Chem. Phys., 19, 3417–3432, https://doi.org/10.5194/acp-19-3417-2019, 2019.

Rieder, H. E., Chiodo, G., Fritzer, J., Wienerroither, C. & Polvani, L. M. Is interactive ozone chemistry important to represent polar cap stratospheric temperature variability in Earth-system models? *Environ. Res. Lett*. **14**, 044026 (2019).

---

## Author Comment (AC1)

**Response to Referee's Comments**

**Manuscript ID: egusphere-2024-2740**

Title: Effects of Ozone-Climate Interactions on the Long-Term Temperature Trend in the Arctic Stratosphere

Author(s): Siyi Zhao, Jiankai Zhang, Zhe Wang, Xufan Xia and Chongyang Zhang

February 2025

**Summary of revision in manuscript**

We sincerely thank the reviewer for your important comments and assistance on our manuscript. The main revisions are summarized as follows:

1. Following the reviewer's suggestions, we created a 1980-clim experiment with an initial field set to fixed boundary conditions for 1980. The climatological mean ozone from the 1980-clim experiment is imported into the radiation scheme of the O3clm experiment. Then, we rerun the O3clm experiment.

2. The reviewers mentioned that only one experiment was unable to exclude the effect of interannual variability in the long-term trend. It was suggested that more ensemble experiments should be conducted. Therefore, we conducted five ensemble experiments for the control and O3clm experiments using different initial fields to ensure the robustness and reliability of the results in the revised manuscript.

3. A detailed description of the experiments has been added to the revised manuscript. Please see method section 2.3: Model and experimental configurations.

4. In order to better align the introduction with the long-term trends in stratospheric ozone and temperature, we have written the introduction

**Response to Comments of Reviewer #1**

I appreciate the authors efforts to isolate the impact of past Arctic ozone changes on trends in temperature and dynamics and to underline their findings with mechanical explanations. Unlike other studies, this study focuses on the extended winter season and might therefore extent our knowledge on the impacts of ozone changes from springtime to the whole winter season. However, I have doubts around the entire experiment setup (especially the way the ozone climatology is calculated) and think that the conducted simulations are not suitable to achieve the goal of this study. Moreover, the authors make use of a figure from another study without mentioning it.

**Response: We are sincerely grateful for the insightful comments and constructive suggestions provided by Reviewer #1. In response to the feedback from Reviewer #1 and the community, we have conducted new ensemble experiments and incorporated supplementary discussions with physical interpretations in relevant sections to enhance the manuscript's clarity. The introduction section has also been strengthened to better emphasize the significance of this study. We apologize for the extended review cycle due to the high computation costs of the ensemble experiments for control and O3clm experiments.**

**Main comment #1:**

More details should be provided on the ozone climatology that is used in the O3clim experiments. Are these daily or monthly means? 3D or zonal mean? If I understand it correctly from Fig. 1, then the ozone chemistry is still calculated in the O3clim setup, but it is not radiatively active. This information is missing in the experiment description in lines 179. How many ensemble members were simulated? Moreover, Fig. 1 is identical to Fig. 1 from Friedel et al. (2022b) and the caption accompanying this figure is almost identical to the caption of Fig. 1 in Friedel et al. (2022a). This must be cited!

**Response: We thank the reviewers for your comments. We have expanded the experimental description in the main text and incorporated their suggestions. In the previously submitted manuscript, we used one ensemble experiment, and for**

the robustness of the experimental results and at the same time to reduce the uncertainty in the effects of ozone-climate interactions, in the revised manuscript we added five ensemble experiments for the new control experiment and the new O3clm experiment. Ensemble members have identical boundary conditions and radiation scheme, but with different initial conditions, which account for the internal variability of the climate system. We have added this specific information in the revised manuscript. Here is revised text as follows (Please see lines 244-261 in the revised manuscript):

"*To understand the causality of the ozone-circulation coupling, we perform model experiments to isolate the impact of ozone changes on stratospheric dynamics and circulation. Two groups of ensemble climate model experiments (i.e., the control experiment and O3clm experiment) use identical boundary conditions and initial conditions. Each group simulation consists of 5 ensemble members, with initial temperature conditions randomly perturbed. Both of the two experiments run from 1970–2020, and the first 10 years are the spin-up time. The control experiment uses fully interactive ozone chemistry, and long-term stratospheric ozone changes are involved in the radiation scheme. In contrast, in the O3clm experiment, the climatological mean ozone is represented by monthly 3-dimensional mean data from a 1980-clim experiment, which is imported into the radiation scheme. In the 1980-clim experiment, surface emissions, external forcing, stratospheric aerosols, fixed lower boundary conditions, and the solar photon enerspectra are all fixed at 1980. The 1980-clim experiment runs for 40 years with the first 10 years as spin-up time and the remaining 30 years of data are used to drive the radiation scheme of the O3clm experiment. This results in the production of fixed radiative feedback, which is to say that the ozone-climate interactions over a long period are not radiatively active. Meantime, this setting is designed to preserve the seasonal temperature variations that conform to the background environmental conditions of the Earth and ensure stable operation of the experiment. Thus, the comparison between the ensemble mean of control and O3clm experiments isolates the feedback effects of*

*long-term stratospheric ozone changes on atmospheric temperature and circulation from climate variability. Figure 1 (adapted from Friedel et al. 2022a, 2022b) provides the inspiration for the experimental design, which is crucial to understanding the analysis presented in this study."*

**In addition, we sincerely apologize for the omission of Figure 1 in the original manuscript, which is similar to Figure 1 in Friedel et al. (2022b). We acknowledge that this figure is similar to Friedel et al. (2022b) and needs to be cited. In the revised manuscript, we have written the experimental design diagram for better understanding of the purpose of the experimental design. Furthermore, we have cited Friedel et al. (2022a; 2022b) to acknowledge their original work and contribution (see lines 264-268):**

*"Figure 1 Simulation setup of the ensemble control and O3clm experiments. The control experiment treats ozone chemistry fully interactively. That is, the calculated ozone field has direct feedback on the atmosphere via the model radiation scheme. In contrast, the ensemble O3clm experiments do not use interactively calculated ozone in the radiation module. Instead, the radiation module uses an ozone climatology, which is derived from the 1980-clim experiment (see Method text) (This figure adapted from Fig. 3a in Friedel et al., 2022a and Fig.1 in Friedel et al., 2022b)."*

[Figure]

**Figure R1. Simulation setup of the ensemble control and O3clm experiments. The control experiment treats ozone chemistry fully interactively. That is, the calculated ozone field has direct feedback on the atmosphere via the model radiation scheme. In contrast, the ensemble O3clm experiments do not use interactively calculated ozone in the radiation module. Instead, the radiation module uses an ozone climatology, which is derived from the 1980-clim experiment (see Method text) (This figure adapted from Fig. 3a in Friedel et al., 2022a and Fig.1 in Friedel et al., 2022b).**

**Main comment #2:**

If I understand the experiment design correctly, I believe that the conducted simulations are not suitable to study the impact of past ozone changes on trends in temperature and dynamics. The authors contrast one simulation with fully interactive ozone for the period 1980-2020 against a simulation where they impose an ozone climatology calculated over this period. Then, they contrast trends from 1980-2000 in the two

simulations to isolate the effect of ozone changes during that period on the variables of interest. I believe that this experiment design does not allow to disentangle the effect of past ozone changes, as the climatology was calculated over a period that includes ozone trends. Therefore, the ozone climatology used in the O3clim run might show lower ozone concentrations (due to an ozone decline from 1980-2000) than the ozone field in the crtl run in 1980. Hence, at the starting point of the simulations (i.e. 1980), the circulation/temperature and ozone field are not consistent in the O3clim simulation. Therefore, temperature changes might arise to adjust to the lower ozone concentrations and trends in this simulation cannot solely be attributed to dynamical and thermodynamical processes other than ozone. Moreover, with the approach used here, effects of potential long-term ozone changes and interannual ozone changes are mixed, since they are both disabled in the O3clim experiments. When only using one ensemble member of 20 years, the role of interannual ozone variability is likely not negligible.

In order to achieve the goal posed by this study, I suggest first deriving an ozone climatology using a time-slice simulation with fixed boundary conditions (i.e. CFCs, GHGs,...) of the year 1980 with fully interactive ozone. The ozone climatology calculated this way can then be used to simulate the O3clim experiments from 1980-2000. Using this approach, the authors would be able to attribute changes in dynamics and temperature over this period to changes in ozone. However, more than one ensemble member for the transient runs would be necessary in order to isolate the effect of longterm ozone changes rather than the effect of interactive ozone variability.

**Response: Thank you for your detailed and thoughtful comments. The reviewer pointed out that the ozone climatology used in the current O3clm simulation may have been influenced by the negative trend in ozone during the 1980−2000 period. In the revised manuscript, we first conducted a 1980 ozone climatology experiment (1980-clim experiment) using 1980 fixed boundary conditions. The detailed information of experimental setup has been given in Major comment #1. The ozone output from the new O3clm experiment can reflect a stable baseline without ozone long-term trends, ensuring consistency between the ozone field in the O3clm**

experiment at the start of the transient run and the circulation and temperature fields. It also removes the potential influence of long-term negative trends on the stratospheric circulations. We compare the climatological ozone from the 1980-clim experiment with the ozone from the control experiment in the original manuscript. The climatological ozone differences between the 1980-clim experiment and the control experiment for 1980−2020 and for different seasons are given in Figure R2. Note that the climatological ozone from the control experiment, that is used to force the old O3clm experiment in the previous manuscript, is lower than that in the 1980-clim experiment, imply a potential effect of negative ozone trend in the old O3clm experiment. Therefore, in the revised manuscript, we use the climatological ozone with seasonal cycle derived from the 1980-clim run as the ozone input field for the new O3clm experiment to exclude the effect of a negative ozone trend.

[Figure]

**Figure R2. Differences in the climatological ozone (unit: ppmv) between the 1980-clim experiment of which climatological ozone forces the new O3clm experiment,and the control experiment of which climatological ozone forces the old O3clm experiment, shown for the annual mean (a), winter mean (December–February) (b), spring mean (March–May) (c), summer mean (June–August) (d), and autumn mean (September–November) (e).**

The new O3clm experiment was also compared with a fully interactive ozone simulation (the control experiment) to accurately isolate the effects of ozone changes on temperature and its long-term trend. The reviewers suggested that the role of interannual ozone change may not be negligible. Therefore, in order to isolate the effects of long-term ozone trend from interannual ozone variability, we increased the number of ensemble members both for the control experiments and the O3clm experiments up to five, reducing the influence of interannual variability in the trend analysis. We have focused on analyzing the effects of long-term ozone changes on the temperature trend using the ensemble mean of the 5 ensemble members.

Figures R3 and R4 give the normalized time series and linear trends in Arctic stratospheric temperatures for different periods (before 2000 and after 2000) from early winter to early spring, with mean values ranging from 10 to 150 hPa. The control experiment in the original manuscript shown in Fig. R3 (hereafter referred to as 'old control experiment) is one ensemble member of the 5 ensemble control experiments, while Fig.4 shows the ensemble mean of the control experiments. A comparison of Fig.3 and Fig.4 (one ensemble member and ensemble mean of 5 members) shows that the added ensemble experiments do not substantially change the Arctic stratospheric temperature trend. From November to December, the Arctic stratospheric temperature from both the MERRA-2 and ERA5 reanalysis

datasets show a small positive trend before 2000 and an insignificant negative trend after 2000 (Figs. R3a, R4a; black line for MERRA-2 and grey line for ERA5). This suggests a warming trend in the Arctic stratosphere during early winter before 2000 and a cooling trend after 2000. The ensemble mean of the control experiments reproduce these opposite trends well, with a significantly positive trend in temperature before 2000 and a significantly negative trend after 2000 (Figs. R3a, R4a; purple line). From January to February, the temperature displays an insignificant negative trend before 2000 in the old control experiment (Fig. R3b) and a significant negative trend after 2000 in the ensemble control experiments (Fig. R4b). From March to April, the temperature from all three datasets shows a significant negative trend before 2000, while there is no significant trend after 2000 in the ensemble mean of the control experiments and the old control experiment (Figs. R3c, R4c). Overall, the long-term trends in temperature derived from the ensemble mean of control experiments are nearly consistent with the results of the reanalysis datasets, both in the period before 2000 and after 2000. In addition, the results of the ensemble mean of control experiments are similar to those of the previous control experiment in the original manuscript, but the ensemble mean of control experiments is able to exclude the internal variability of the climate system; therefore, we use the ensemble mean of the control experiments in the revised manuscript.

[Figure]

Figure R3. Normalized time series of the temperature averaged from 150 hPa to 10 hPa over 65°–90°N from 1980–2020 in (a) November–December, (b) January–February, and (c) March–April derived from the MERRA-2 (red column), ERA5

(orange column) reanalysis dataset and CESM control experiments used in the original manuscript (brown column). The color straight lines represent the linear trends during the pre-2000 and post-2000 periods. Solid lines indicate that the trends are statistically significant at the 90% confidence level according to Student's *t* test. This is Figure 2 in the original manuscript.

[Figure]

Figure R4. Same as Figure R3, but derived from the ensemble mean of the control experiments. This is Figure 2 in the revised manuscript.

Figure R5 shows the trends in daily temperature and ozone between 10 and 250 hPa in the polar cap regions (65°–90°N) during the pre-2000 period, which are based on data from MERRA-2 and the control experiments. Fig. R5c, R5d shows the results derived from the old control experiment in the original manuscript, while Fig. R5e, R5f shows the results of the ensemble mean of the control experiments. The trend reversal is also evident in December, which is consistent with Figs. R3 and R4. During November and December, there is an increasing trend in both temperature and ozone across all levels in the old control experiment (Fig. R5a, R5b). While after December, the trends in temperature and ozone reverse in the middle stratosphere and then in the lower stratosphere. Similar trend patterns are found in the old control experiment (Fig. R5c, R5d) and the ensemble control experiments (Fig. R5e, R5f), indicating that the ensemble control experiments can reproduce the long-term trends in stratospheric temperature and ozone in both early and late winter. In summary, the additional ensemble members do not affect the trends in Arctic stratospheric temperature.

[Figure]

**Figure R5. Time evolution of trends in daily temperature (a, c and e) and ozone (b, d and f) in the polar cap regions (65°–90°N) during winter and spring derived from MERRA2 (a, b), the one ensemble of control experiment (c, d) and the 5 ensemble mean of the control experiments (e, f) in the pre-2000 period. The green dotted regions indicate that the trends are statistically significant at the 90% confidence level according to Student's *t* test. This is Figure 3 in the original manuscript.**

**Figure R6a, R6b show the daily trends in temperature and ozone between 10 and 250 hPa in the polar cap regions (65°–90°N) before 2000 derived from the old O3clm experiment. The old O3clm experiment shows a nonsignificant temperature trend from November to December and a positive trend in late February and March. This result is opposite to those in the old control experiment (Fig. R5b, R5c). The stratospheric ozone exhibits negative trends over the 10–100**

hPa from November to April, without an intra-seasonal reverse. The differences in trends of temperature and ozone between the old control experiment and old O3clm experiment are shown in Fig. R6c and R6d, which also shows an intra-seasonal reverse around December.

Figure R7a and R7b show similar trends to those in Figure R6a and R6b, except that the non-significant positive temperature trend from late February to early March became smaller and shorter in duration. The negative temperature trend in April is larger than those in the original manuscript but still insignificant. The ozone trend in the lower stratosphere is different from the previous results, as evidenced by a non-significant positive trend below 30 hPa in December and a negative ozone trend during January-April. It is worth noting that the ozone trend in late winter and spring derived from the new ensemble O3clm experiments still shows significant negative trends, which may be contributed by the chemical ozone depletion induced by increasing ODSs in the pre-2000 period (Fig. R7b). Conversely, the stratospheric ozone during February and early spring in the new O3clm experiment in the post-2000 period shows a significant positive trend (Fig. 13b in the revised manuscript), which is consistent with that in the old O3clm experiment in the original manuscript. This result suggests that the new O3clm experiment has excluded the potential impact of ozone decline as input field, which is the biggest concern of the reviewer #1. In addition, the difference in trends between the ensemble mean of the control and O3clm experiments is basically consistent with those in the original manuscript, except that the trend is smaller compared to the previous results, which is partially contributed by internal variability. These results indicate that the new control and new O3clm experiments with increased 5 ensemble members do not substantially affect the analysis results of the ozone-climate interaction feedbacks, although there is a weaker reduction in the strength of the trend in Arctic stratospheric temperature.

[Figure]

**Figure R6. Time evolution of the trend of daily temperature and ozone over the levels between 10 and 250 hPa in the polar cap regions (65°–90°N) during winter and spring derived from the O3clm experiment (a–b) and the differences between the control experiment and O3clm experiment (c–d) before 2000. The green dotted regions indicate that the trend is statistically significant at the 90% confidence level according to Student's *t* test. This is Figure 4 in the original manuscript.**

[Figure]

**Figure R7. Same as Figure R6, but from the 5 ensemble mean of the O3clm**

experiments and the difference between the ensemble mean of the control and O3clm experiments. This is Figure 4 in the revised manuscript.

The core process of ozone-climate interactions is ozone-circulation feedback. Figure R8 displays the trend in the vertical component downwelling branch of the BDC ($\overline{w}^*$) averaged over the polar regions (65°–90°N) during the pre-2000 period in the ensemble control experiments and O3clm experiments. We decomposed these trends into contributions from wave 1 (Fig. R8b, e and h) and wave 2 (Fig. R8c, f and i). The ensemble mean of the control experiments shows significant negative trends in $\overline{w}^*$ from November to early December, corresponding to enhanced downwelling compared to climatological mean, and positive trends in $\overline{w}^*$ from late December to January, corresponding to weakened downwelling (Fig. R8a). After February, the trends in $\overline{w}^*$ are less significant (Fig. R8a). The linear trends in $\overline{w}^*$ are basically opposite to those in temperature derived from the ensemble control experiments (Fig. R5e), which is because the enhanced downwelling (upwelling) favors polar adiabatic warming (cooling). Additionally, the $\overline{w}^*$ trend contributed by wave 1 is similar to the total trend, suggesting that wave 1 dominates the trends in $\overline{w}^*$. In the ensemble O3clm experiments, there is no negative trend in $\overline{w}^*$ in November and early December (Fig. R8d–f). This result indicates that ozone-circulation feedback strengthens the $\overline{w}^*$ in early winter, leading to adiabatic warming in early winter; conversely, there are anomalous upward motions that induce anomalous adiabatic cooling from January to February, which is consistent with the reversal of the temperature trend in January (Figs. R5, R7). The difference between the ensemble control and O3clm experiments suggests a similar pattern to that of the ensemble control experiments (Fig. R8g, h and i). The results from the ensemble mean of the control

experiments, O3clm experiments and the difference between the ensemble mean of the two experiments are consistent with those from the old experiments, indicating that the Arctic stratospheric temperature trends are indeed driven by ozone-climate interactions rather than internal variability. Overall, the results of the new ensemble experiments still support the intra-seasonal reverse in the trend in Arctic stratospheric temperature around early winter.

[Figure]

**Figure R8. Linear trend of the vertical component ($\overline{w}{}^{*}$) of the BDC and its contribution to the wavenumber 1 (b, e and h) and wavenumber 2 components (c, f and i) before 2000 averaged in the polar regions (65°–90°N) during winter and spring, derived from the ensemble control experiments (a, b and c), O3clm experiments (d, e and f) and the differences between the ensemble mean of the control and O3clm experiments (g, h and i). The green stippled regions indicate the trend in $\overline{w}{}^{*}$ significant at the 90% confidence level according to Student's _t_ test (The daily data are first processed with a 30-day low-pass filter to remove high-frequency signals). This is Figure 5 in the revised manuscript.**

Our new experiments also support the finding in the original manuscript that

[revised manuscript text omitted]

In addition, negative temperature anomalies (Figs. R4c, R5c and R5e) emit less longwave radiation. The temperature then increases and generates positive QRL anomalies in spring. The differences in QRS and QRL trends between the ensemble mean of control and O3clm experiments exhibit nearly same patterns as those in the original manuscript. Our results demonstrate that the ozone-climate interactions during early winter mainly influence stratospheric temperature through dynamic adjustments. In contrast, the trends in temperature during late winter and spring are primarily due to dynamic cooling and shortwave cooling. Overall, the results of the ensemble mean of the two experiments do not change the trend of QRL and QRS, which is consistent with the original manuscript.

[Figure]

**Figure R10.** Time evolution of trends in the daily shortwave heating rate (solar heating rate; QRS) and longwave heating rate (QRL) between 10 and 250 hPa in the polar regions (65°–90°N) during winter and spring derived from the ensemble mean of the control experiments (a, b), O3clm experiments (c, d) and the difference between the two experiments (e, f) before 2000. The green dotted regions indicate that the trends are statistically significant at the 90% confidence level according to Student's *t* test (Nine-point smoothing was performed during drawing to remove noisy signals). This is Figure 11 in the revised manuscript.

In summary, the new ensemble mean of the control experiments, O3clm experiments and the difference between the two experiments are consistent with those in the original manuscript, and a detailed explanation of the mechanism is given in Section 4 of the revised manuscript. Although we use a new initial field for the ozone climate state in the O3clm experiment, the effect of the ozone-climate interaction does not change substantially, practically noted that there still exists an intra-seasonal temperature reversal around early winter. Therefore, we

**conclude that the long-term trend in temperature and ozone is not affected by the initial field of the climate state. By redesigning the experiments and adding transient runs with 5 ensemble members, we have effectively addressed the issues raised by the reviewers and significantly improved the scientific validity and reliability of this study.**

**Other comments:**

The introduction does not follow a clear outline. Sometimes it is unclear whether the authors are referring to interannual or long-term processes. The introduction reviews mostly literature that focuses on ozone-climate coupling on interannual timescales, but then the study focuses on long-term trends. Please tailor the introduction a bit better to the aim of this study. Some of the statements made in the introduction are also not supported by the cited literature (see detailed comments below). The title of the study is also misleading, as is suggests that this study focuses on interannual variability rather than long-term trends.

**Response: We thank the reviewers for their constructive comments on the introduction and title of the manuscript. The title of the revised paper has been changed into "*Effects of Ozone-Climate Interactions on the Long-Term Temperature Trend in the Arctic Stratosphere*". Additionally, in order to better align the introduction with the long-term trends in stratospheric ozone and temperature, we have written the introduction and the following is the revised introduction (see lines 23-143):**

[revised manuscript text omitted]

Line 8: "increased"

**Response: Corrected, thank you.**

Line 15: "cooling trends in the Arctic stratosphere are ..."

**Response: Corrected, thank you.**

Line 35: Please be a bit more precise here. Marsh et al. (2016) conclude that ozone feedbacks are not crucial for the model's climate sensitivity in 4xCO2 forcing experiments. Maybe it is worth citing Rieder et al. (2019) in this context, who showed that ozone variations are important for the stratospheric temperature variability in models.

**Response: Thank you for your suggestion. We revised the text and provided additional context on the importance of ozone variations. The revised manuscript as follows (see lines 37-43):**

"*Additionally, Rieder et al. (2019) demonstrated that ozone-climate interactions are important for accurately capturing stratospheric temperature variability in models. However, some studies, such as Marsh et al. (2016), suggested that ozone-climate interactions have limited influences (approximately 1%) on climate sensitivity.*"

Lines 44: I could not find any evidence for this statement (i.e. the longwave ozone effect on stratospheric dynamics during winter) in the cited study (Strahan et al., 2013). Could you please provide further literature on the mid-winter longwave radiative effect of ozone?

**Response: We appreciate the reviewer pointing out this oversight. The revised sentence will read (see lines 47-61):**

*"In winter, although solar radiation in the Arctic regions is absent, ozone can still absorb and emit longwave radiation. Seppälä et al. (2025) pointed out that a reduction in stratospheric ozone could directly lead to stratospheric warming. This longwave radiative warming may influence the strength of the Arctic polar vortex (Hu et al., 2015), further modulating the transport of ozone-rich air from mid-latitudes to the Arctic polar regions (Zhang et al., 2017). In addition, the Arctic ozone can modulate the planetary wave activity, which further influences Arctic stratospheric temperature via wave-mean flow interactions in winter (Nathan and Cordero, 2007; Albers and Nathan, 2013; Hu et al., 2015). Thus, Arctic stratospheric ozone affects stratospheric temperatures through its longwave radiative effects and dynamical processes during winter."*

Lines 47: Are you now talking about mid- or late-winter/spring? I believe this statement is only valid for springtime, when the polar vortex is already weakened. See also Haase and Matthes (2019).

**Response: Thank you for highlighting this point. To address this, we revise the text for clarity and include a reference of Haase and Matthes (2019). The revised sentence is specified as (see lines 61-75):**

*"In late-February and early spring, as solar radiation reaches high latitudes, the polar regions become warm compared to winter and the stratospheric polar vortex is weakened. However, from the perspective of climate, the increase in ozone depleting substances (ODSs) in the 20th century leads to springtime stratospheric ozone depletion and decreased absorption of shortwave radiation, which cools the Arctic stratosphere and strengthens the polar vortex (Friedel et al., 2022a). This results in reduced wave propagation towards the lower stratosphere and thereby a colder Arctic stratosphere (Coy et al., 1997; Albers and Nathan, 2013; Haase and Matthes, 2019). On the one hand, the strengthened Arctic polar vortex decreases ozone transport to the polar regions, further reducing ozone concentrations. On the other hand, a colder Arctic stratosphere facilitates the formation of polar stratospheric clouds (PSCs).*

*PSCs provide sites for heterogeneous reactions. The reactions convert stable chlorine reservoir species into active chlorine, then catalytically destroys ozone (Solomon et al., 1986; Feng et al., 2005a, 2005b; Calvo et al., 2015)."*

Line 85: I do not really understand this sentence (starting with "Lin et al. ... "). Could you please reformulate?

**Response: In the revised manuscript we reorganized the introduction to focus more on the long-term trends in Arctic stratospheric temperature and the deletion of this sentence.**

Lines 91 : Chiodo et al. (2023) studied the impact of long-term ozone trends in the Arctic on temperature and dynamics. Might be worth mentioning this here.

**Response: Thank you for your comment. We have rephrased as follows (see lines 126-127):**

**"*A recent work by Chiodo et al. (2023) has explored the impact of long-term ozone trends on the temperature in the Arctic, providing valuable insights into the ozone-climate interactions.*"**

(9)Lines 94 . "..., we focus on the historical long-term trends ..."
**Response: Corrected, thank you.**

Lines 202: To support this statement on the persistent cooling, you would have to calculate a trend over the whole time period. When doing this, you would probably get no trend, e.g. no significant temperature changes from 1980-2020.

**Response: Thank you for your attention to detail. There is no evidence of sustained cooling, so we modified the expression as follows (see lines 280-282):**

**"*From January to February, the temperature displays an insignificant negative trend*"**

*before 2000 and a significant negative trend after 2000, derived from the three datasets (Fig. 2b)."*

Fig. 2: How have the time series been normalised?

**Response: In this study, the normalized time series are standardized using Z-score standardization, where the data are processed using the following formula:** $A_{s\text{-}value} = \dfrac{A_{o\text{-}value} - \bar{A}}{\sigma_A}$ **, where** $A_{s\text{-}value}$ **denotes the normalized A-value,** $A_{o\text{-}value}$ **denotes original A-value,** $\bar{A}$ **denotes average A-value,** $\sigma_A$ **denotes standard deviation. This method transforms the data into a distribution with a mean of 0 and a standard deviation of 1, making different datasets comparable. Using the Z-score standardization method can better compare trends across different datasets. We added the normalized method information in the section Method and please see lines 231-233.**

The trends in Fig. 2 look a bit constructed, i.e. they seem to be very sensitive to the time chosen time frame. Especially the trends in springtime (March/April) seem to be entirely caused by the ozone depletion event in 1997.

**Response: Thank you for your comment. To rule out the excessive impact of the year 1997, we recalculated the long-term trend by excluding the 1997 data from the calculation of the 2000 trend in Figure R11. The results show that the temperature and ozone trends during the pre-2000 period are not affected by the extreme ozone depletion event, and including 1997 does not change the negative springtime temperature and ozone trends (Fig. R3).**

[Figure]

**Figure R11. Normalized time series of the temperature averaged from 150 to 10 hPa over 65°–90°N from 1980–2020 in (a) November–December, (b) January–February, and (c) March–April derived from MERRA2 (red column), ERA5 (orange column) and CESM (brown column) the ensemble control experiments. The color straight lines represent the linear trends before 2000, which excludes data for 1997, and after 2000. Solid lines indicate that the trends are statistically significant at the 90% confidence level according to Student's *t* test.**

**In addition, the selection of 2000 is based on several scientific considerations: (1) Several studies provide evidence for a turning point in long-term stratospheric trends around the year 2000. Satellite observations from TOMS, OMI, and GOME reveal a turnaround in stratospheric ozone trends around 2000, with stabilization and signs of recovery (WMO, 2018; LOTUS, 2019). Ozonesonde measurements clearly indicate increased lower stratospheric ozone levels after 2000 corroborating satellite findings (IPCC, AR5, 2013). Perlwitz et al. (2008) indicated that the modeled climate response to ozone recovery is almost opposite to that of ozone depletion before 2000. These evidences point to a consensus on the year 2000 as a turning point in ozone depletion and recovery. (2) By dividing the analysis into two periods, pre- and post-2000, we can better isolate the different impacts of ozone depletion and recovery on Arctic stratospheric temperature and dynamics. While it is true that the turning point around 2000 may affect the significance of the trends, this effect is precisely one of the focuses of our study. By comparing trends before and after 2000, we aim to better understand how ozone-climate interactions evolve during the ozone depletion and recovery period.**

---

## Author Comment (AC2)

**Response to Community's Comments**

**Manuscript ID: egusphere-2024-2740**

**Title: Effects of Ozone-Climate Interactions on the Long-Term Temperature Trend in the Arctic Stratosphere**

**Author(s):** Siyi Zhao, Jiankai Zhang, Zhe Wang, Xufan Xia and Chongyang Zhang

**February 2025**

**Summary of revision in manuscript**

We sincerely thank the reviewer for your important comments and assistance on our manuscript. The main revisions are summarized as follows:

1. The reviewers mentioned that only one experiment was unable to exclude the effect of interannual variability in the long-term trend. It was suggested that more ensemble experiments should be conducted. Therefore, we conducted five ensemble experiments for the control and O3clm experiments using different initial fields to ensure the robustness and reliability of the results in the revised manuscript.

2. We provided a detailed response to examine how circulation changes affect Arctic stratospheric ozone's trend. Especially, whether the Brewer-Dobson Circulation (BDC) drive early winter ozone trends.

3. We investigated the role of ozone increase and ozone–circulation interactions in the reversal of the refractive index (RI) from November to December.

4. Some sentences have been rewritten as well as the grammar is improved throughout the manuscript.

**Response to Comments of Community**

The Arctic lower stratosphere experienced a warming trend in early winter but a cooling trend in late winter/early spring during 1980-1999. During the same period, the Arctic lower stratospheric ozone increased in early winter and decreased in late winter. This paper investigates the effects of stratospheric ozone changes on Arctic lower stratospheric temperature trends using CESM simulations and reanalysis. It is found that the early winter Arctic lower stratospheric warming was caused by enhanced dynamical warming, which was strongly modulated by the increase of Arctic ozone. In late winter/early spring, Arctic ozone depletion reduces shortwave heating and causes lower stratospheric cooling.

Overall, the paper is well written. The results can improve our understanding of how Arctic ozone change affects climate, which has received less attention. However, I have some major concerns about the analysis and I think a major revision is needed.

**Response: We sincerely appreciate your thorough feedback and valuable suggestions. In response to Reviewer #1 and your comments, we have incorporated comprehensive new experiments, expanded discussions, and enhanced physical interpretations throughout relevant sections of the manuscript. These substantive revisions aim to strengthen the theoretical framework and improve the manuscript's clarity and scientific validity. We apologize for the extended review cycle due to the high computation costs of the ensemble experiments for control and O3clm experiments.**

**Major comment #1:**

The authors have done a detailed analysis of how Arctic stratospheric ozone changes affect the circulation, but they do not investigate how circulation changes affect Arctic ozone. For example, ozone increase in early winter leads to stronger wave propagation into the stratosphere and Arctic warming. They should also consider the effects of an enhanced BDC on Arctic ozone increase. Indeed, the increasing ozone trend in early winter must be driven by dynamics.

**Response: We sincerely thank the reviewers for their thorough review and valuable feedback. We agree with the reviewers that it is important to examine how circulation changes affect Arctic stratospheric ozone's trend. Especially, whether the enhanced wave propagation and the Brewer-Dobson Circulation (BDC) drive early winter ozone trends. We provide a detailed response to the comment and outline revisions made to address the issue.**

**Previous studies used Transformed Eulerian Mean (TEM) equation combined with zonal-mean ozone tracer continuity equation to diagnose the ozone transport induced by the Brewer-Dobson circulation (BDC) and ozone eddy transport (Monier and Weare, 2011; Abalos et al., 2013; Zhang et al., 2017). The ozone budget equations are represented as follows (Monier and Weare 2011; Abalos et al. 2013):**

$$\frac{\partial \overline{\chi_{O_3}}}{\partial t} = \frac{\overline{v}^*}{R}\frac{\partial \overline{\chi_{O_3}}}{\partial \phi} - \overline{w}^*\frac{\partial \overline{\chi_{O_3}}}{\partial z}\,(term1)$$
$$-\frac{1}{\rho_0}\nabla \cdot M\,(term2) \qquad \textbf{(Eq.1)}$$
$$+\overline{S}\,(term3)$$

**where $\overline{S}$ is the sum of all chemical sources and sinks, $\overline{\chi}_{O_3}$ is the zonal-mean ozone concentration, $\overline{v}^*$ and $\overline{w}^*$ are the meridional and vertical BDC velocities (Andrews et al. 1987), respectively; $M$ is the eddy flux vector, which is represented as:**

$$\left[ \rho_0\left( \overline{v'\chi'_{O_3}} - \frac{\overline{v'\theta'}}{\overline{\theta}_z}\frac{\partial \overline{\chi_{O_3}}}{\partial z} \right), \rho_0\left( \overline{w'\chi'_{O_3}} + \frac{1}{R}\frac{\overline{v'\theta'}}{\overline{\theta}_z}\frac{\partial \overline{\chi_{O_3}}}{\partial \phi} \right) \right], \quad \textbf{(Eq.2)}$$

**$\nabla \cdot M$ is the divergence of the eddy flux vector and represents the eddy transport of ozone; $\rho_0$ is air density; $\theta$ is potential temperature; $R$ is Earth's radius; $t$ is time; $\phi$ and $z$ are latitude and height, respectively.**

Figure R1 shows the trend in stratospheric ozone budget from November to February between 10 and 250 hPa in the polar regions (65°–90°N) in the pre-2000 period, which is decomposed into BDC and eddy transport of ozone (term1 and term2 in Eqs. (1)). In the ensemble control experiments, from November to December (early winter), the total ozone budget shows a significantly positive trend, indicating an increase in ozone concentrations. This trend is primarily driven by the sum of BDC and eddy transport. In mid-winter, the trend in ozone budget weakens and changes to negative, indicating a leveling off of increased ozone concentration. In contrast, in the ensemble O3clm experiments, the trend in the ozone budget is opposite to those in the ensemble control experiments and is not statistically significant from November to February. This demonstrates that during early winter, the accelerated BDC intensifies poleward ozone advection through directly transports ozone-rich air masses from tropical reservoirs to polar region, and enhances downward transport of ozone from the upper stratosphere to lower stratosphere. The transport of ozone due to ozone-circulation feedback is reconfirmed by the difference between the ensemble mean of the control and O3clm experiments. In January, the difference between the two experiments shows an intra-seasonal reverse in ozone transport, indicating that the ozone-circulation interactions can also give feedback to ozone concentrations.

[Figure]

**Figure R1. Dynamically produced ozone concentration trend, decomposed into (a, d and g) meridional and (b, e and h) vertical BDC transport and (c, f and i) eddy transport between 10–150 hPa in the polar regions (65°–90°N) from November to February, derived from (a–c) the ensemble control, (d–f) the O3clm experiments and (g–i) the difference between the two experiments during the pre-2000 period. The trend over the dotted regions is statistically significant at the 90% confidence level according to the Student's *t* test (The daily data are first processed with a 30-day low-pass filter to remove high-frequency signals).**

**The abovementioned analysis has been added in the revised paper and please see lines 375-397 and Figure 6 in the revised manuscript.**

**Major comment #2:**

Line 315-317. This is a key result of this study. Note that Arctic lower stratospheric ozone has an increasing trend in Nov-Dec in the control experiment (Fig. 3d). Please explain how ozone increase (or ozone-circulation interactions) leads to a reversal of the

refractive index from November to December.

**Response: Thanks a lot for your comments. Nathan and Cordero (2007) pointed that wave-induced ozone heating decrease wave drag by about 25% in the lower stratosphere, favoring planetary wave propagation at this altitude during early winter in the present study (Figure R2; Figure 7a, g in the revised manuscript). Additionally, they pointed out that photochemically accelerated cooling due to ozone augments the Newtonian cooling and increases the wave drag by a factor of two in the upper stratosphere, which is in accordance with our finding that ozone-climate interactions enhance the upper stratospheric E-P flux convergence (Figure 7a, g in the revised manuscript). These analysis results highlight how ozone-climate interactions affect stratospheric dynamics processes.**

**Here we used wave refractive index (RI) change to analyze the influence of ozone-climate interactions on wave propagation. RI is used to diagnose the environment of wave propagation (Chen and Robinson, 1992) and is calculated as:**

$$RI = \frac{\overline{q}_{\varphi}}{\overline{u}} - \left(\frac{k}{a\cos\varphi}\right)^2 - \left(\frac{f}{2NH}\right)^2 \quad \textbf{(Eq.3)}$$

**where the meridional gradient of the zonal mean potential vorticity is calculated as:**

$$\overline{q}_{\varphi} = \frac{2\Omega}{a}\cos\varphi - \frac{1}{a^2}[\frac{(\overline{u}\cos\varphi)_{\varphi}}{a\cos\varphi}]_{\varphi} - \frac{f^2}{\rho_0}(\rho_0 \frac{\overline{u}_z}{N^2})_z \quad \textbf{(Eq.4)}$$

**where** $-\frac{f^2}{\rho_0}\left(\rho_0 \frac{\overline{u}_z}{N^2}\right)_z = \left(\frac{f^2}{HN^2} + \frac{f^2}{N^4}\frac{\mathrm{d}N^2}{\mathrm{d}z}\right)\overline{u}_z - \frac{f^2}{N^2}\overline{u}_{zz}$ **, and** $H, q, k, N^2, \Omega, u_z$ **are the scale height, potential vorticity, zonal wavenumber, buoyancy frequency, Earth's angular frequency, and zonal wind shear, respectively. Note that the second term of RI does not change with atmospheric state, which is always positive, and the third term of RI is insignificant compared to the first term. The second**

term is also insignificant for planetary waves with very small wave numbers (Hu et al., 2019). Previous studies indicate that changes in zonal mean potential vorticity meridional gradient $\bar{q}_\varphi$ could explain the most of changes in RI in the middle and high latitudes (e.g., Hu et al., 2019; Simpson et al., 2009).

Figure R2 shows the daily evolution of the trend in the RI, the vertical component of the E-P flux ($F_z$ )and $\bar{q}_\varphi$ averaged between 45°–75°N and U60 (zonal wind at 60°N) in the lower stratosphere (50–150 hPa) before 2000. The datasets are derived from the ensemble control experiments and O3clm experiments. Specifically, in the ensemble control experiments, positive zonal wind vertical shear anomalies (Fig. R5) at middle latitudes during November increase the $\bar{q}_\varphi$ (Fig. R3), which in turn raises the RI and enhances the $F_z$ (Fig. R2; purple lines). The increase in planetary waves in early winter weakens the polar vortex compared to that in the O3clm experiment, leading to deceleration in circumpolar westerlies during mid-December and January (red lines in Fig. R2). The decreased zonal wind around 60°N further suppresses the vertical propagation of planetary wave in the subsequent winter months, corresponding to the intra-seasonal reversal of $F_z$ before and after January. Then, the weakening of $F_z$ in the ensemble control experiments allows for a stronger recovery of the polar vortex due to wave-flow interaction in February compared to the O3clm experiments (red lines in Fig. R2). This intra-seasonal reversal of $F_z$ explains the reversals of BDC and temperature around December, and this feature disappears in the ensemble O3clm experiments in which the ozone-interactions are cut off, highlighting the key role of ozone-climate interactions in modulating stratospheric dynamics processes. We have added the above-mentioned results in the revised manuscript (Please see lines 468-484 and Figure 8).

[Figure]

**Figure R2. Daily evolution of the trends in the RI (black lines), vertical component of the E-P flux ($F_z$; purple lines), $\bar{q}_\varphi$ (blue lines), U60 (zonal wind at 60°N; red lines) before 2000 at 50–150 hPa averaged in mid-latitude (45°–75° N) from 1 November to 28 February, derived from (a) the ensemble mean of the control experiments and (b) O3clm experiments. The solid lines indicate the trends in the significant RI, vertical component of the E-P flux and $\bar{q}_\varphi$ at the 90% confidence level according to Student's *t* test (The daily data are first processed with a 7-day low-pass filter to remove high-frequency signals).**

We further analyzed which term dominates the change in $\bar{q}_\varphi$. Figure R3 shows the pattern of the difference in $\bar{q}_\varphi$ between the high ozone period (1980−1985) and the low ozone period (1997−2002). According to the Eq. (4), the first term of $\bar{q}_\varphi$ does not change with the atmospheric state. Therefore, the second term

$(-[\dfrac{(\bar{u}\cos\varphi)_\varphi}{a\cos\varphi}]_\varphi$ ; hereafter referred to as the $U_{yy}$ term or barotropic term) and the

third term $(-\dfrac{f^2}{\rho_0}(\rho_0\dfrac{\bar{u}_z}{N^2})_z$ ; hereafter referred to as the $U_{zz}$ term or baroclinic term) are investigated. In the ensemble control experiments, the pattern of responses in the $U_{zz}$ term is similar with $\bar{q}_\varphi$ (Figs. R3, R5). This implies that changes in $\bar{q}_\varphi$ over the Arctic in the stratosphere are mainly due to the $U_{zz}$ term. The baroclinic term plays a dominant role in modulating the $\bar{q}_\varphi$ in the Arctic stratosphere. Similar results were obtained in a study from Hu et al. (2022). A reversal of $\bar{q}_\varphi$ in the upper troposphere and lower stratosphere (UTLS) over 60–70°N (Fig. R3a) leads to a reversal of the $\bar{q}_\varphi$ and RI from November to December. In the ensemble O3clm experiments, there is no significant $\bar{q}_\varphi$ increase in the UTLS region in November, nor did the baroclinic term provide favorable conditions (Fig. R5d). This suggests that ozone-climate interaction promotes planetary wave upward by affecting the baroclinic term, which in turn induces an increase in RI.

[Figure]

Figure R3. Altitude-latitude cross-section of difference in $a^2\cdot\bar{q}_\varphi$ between the high

ozone period (1980−1985) and the low ozone period (1997−2002), derived from the ensemble control experiments (a, b and c) and O3clm experiments (d, e and f). Green dots indicate that the differences are statistically significant at the 90% confidence level according to Student's *t*-test.

[Figure]

**Figure R4. Same as Figure R3, but for the** $-[\dfrac{(\bar{u}\cos\varphi)_{\varphi}}{a\cos\varphi}]_{\varphi}$ **(U$_{yy}$ term or barotropic).**

[Figure]

**Figure R5. Same as Figure R3, but for the** $-a^2 \cdot \dfrac{f^2}{\rho_0}(\rho_0 \dfrac{\bar{u}_z}{N^2})_z$ **(U$_{zz}$ term or baroclinic).**

**Minor Comments:**

Lines 9-12: The paper does not show any result of ozone-induced longwave cooling. So, it should not be included in the abstract.

**Response: Thank you for your comment. Actually, in our previous manuscript, Fig. 11f shows the result of ozone-induced longwave cooling. We described in detail the changes in longwave cooling induced by ozone and its effect on temperature in the main text (Figure 11 and lines 544-569 in the revised manuscript).**

Line 15: "enhanced shortwave radiative cooling" --- reduced shortwave radiation warming

**Response: Thank you for pointing this out. We rectified this expression in the revised manuscript, as detailed in lines 14-16:**

**"*In contrast, during late winter and spring, cooling trends in the Arctic stratosphere are predominantly driven by the reduced shortwave radiation heating associated with stratospheric ozone depletion.*"**

Line 180: Is there only one member for each experiment?

**Response: Thank you for your comments. In the original manuscript, each experiment includes only one ensemble member. We acknowledge that having more ensemble members would improve the reliability of the results by averaging out random interannual variability. Therefore, in the revised manuscript, we used five ensemble members both in control experiment and O3clm experiment to**

**reduce the experimental uncertainties. The analysis of new results derived from ensemble experiments can be found in Main comment #2 of the reviewer 1.**

Lines 190-193: I wonder why you want to calculate ozone interactively in the O3clm experiment since the calculated ozone is not used in radiation. Why not just prescribe ozone?

**Response: Thanks for your comment. The decision to calculate ozone interactively in the O3clm experiment, even though it is not used in the radiation scheme, is motivated by the need to maintain consistency in the model's chemical and dynamical processes. By allowing the ozone to be calculated interactively, the model ensures that the chemical processes involving ozone (such as its production and destruction) are consistent with the rest of the atmospheric chemistry. This is important because ozone interacts with other chemical species, and these interactions can influence the overall atmospheric state. Previous studies suggested that climate models without chemical-radiative-dynamical feedback process cannot capture the realistic variability of stratospheric compositions and other stratospheric processes (Cionni et al., 2011; Eyring et al., 2013; Jones et al., 2011). In addition, prescribing ozone might introduce biases in the model, especially if the prescribed ozone fields do not perfectly match the model's internal state. By calculating ozone interactively, the model avoids potential discrepancies that could arise from using prescribed fields, ensuring a more self-consistent simulation. Notably, the primary goal of the O3clm experiment is to isolate the effects of ozone-climate interactions by comparing it with the control experiment where ozone is fully interactive, especially for the chemical-radiative-dynamical processes induced by long-term ozone changes, which is our main point of innovation.**

Lines 209-211: Move the two sentences to the beginning of the paragraph.

**Response: Corrected, thank you.**

Line 220: What causes the lower stratospheric ozone increase in Nov-Dec?

**Response: The increase in lower stratospheric ozone during November–December is primarily driven by dynamical processes. Specifically, the enhanced ozone transport is induced by the BDC and eddy transport. Key contributing factors as follows: (1) Enhanced ozone transport by the BDC: In early winter, planetary wave activity leads to increased upward and poleward transport of ozone-rich air from the tropics into the Arctic lower stratosphere. This dynamical transport is particularly pronounced during years with strong wave driving, which accelerates the BDC's downwelling branch and leads to ozone accumulation in the Arctic lower stratosphere. (2) Ozone eddy transport: Eddy transport of ozone transports ozone-rich air into the Arctic lower stratosphere. (3) Suppressed ozone loss: During early winter, an absent of solar radiation levels reduce the activation of catalytic ozone-destroying reactions involving halogens. This radiative condition allows transported ozone to accumulate with minimal chemical loss. The updated text as follows (see lines 375-397):**

"*Furthermore, the enhanced BDC may have an effect on the ozone concentration. The increase in stratospheric ozone during November–December and decrease during January–February (Fig. 4d) induced by ozone-circulation feedback is caused by enhanced dynamical transport. We focus on the role of the BDC in driving the ozone increase in early-winter and its decrease in mid-winter, investigating the reasons for the reversal. Figure 6 shows the trend in stratospheric ozone budget from November to February between 10 and 250 hPa in the polar regions (65°–90°N) in the pre-2000 period, which is decomposed into BDC and eddy transport of ozone (calculated by Eqs. (11), (12)). In the ensemble control experiments, from November to December (early winter), the total ozone budget shows a significantly positive trend, indicating an increase in ozone concentrations. This trend is primarily driven by the*

*sum of BDC and eddy transport. In mid-winter, the trend in ozone budget weakens and changes to negative, indicating a leveling off of increased ozone concentration. In contrast, in the ensemble O3clm experiments, the trend in the ozone budget is opposite to those in the ensemble control experiments and is not statistically significant from November to February. This demonstrates that during early winter, the accelerated BDC intensifies poleward ozone advection through directly transports ozone-rich air masses from tropical reservoirs to polar region, and enhances downward transport of ozone from the upper stratosphere to lower stratosphere. The transport of ozone due to ozone-circulation feedback is reconfirmed by the difference between the ensemble mean of the control and O3clm experiments. In January, the difference between the two experiments shows an intra-seasonal reverse in ozone transport, indicating that the ozone-circulation interactions can also give feedback to ozone concentrations."*

**For a detailed explanation, please see our replies for your major comment#2.**

Line 222: "observed" --- found

**Response: Corrected, thank you.**

Lines 301-310: Which term in equation (5) causes the reversal of the PV gradient? Also see my major comment 2.

**Response: Thank you for your comment. We futher analyze which term dominates the change in $\bar{q}_\varphi$. Figure R3 shows the pattern of the difference in $\bar{q}_\varphi$ between the high ozone period (1980−1985) and the low ozone period (1997−2002). According to the Eq. (4), the first term of $\bar{q}_\varphi$ does not change with the atmospheric state. Therefore, the second term ($-[\frac{(\bar{u}\cos\varphi)_\varphi}{a\cos\varphi}]_\varphi$; hereafter referred to as the $U_{yy}$ term**

or barotropic term) and the third term ($-\dfrac{f^2}{\rho_0}(\rho_0 \dfrac{\bar{u}_z}{N^2})_z$; hereafter referred to as the

$U_{zz}$ term or baroclinic term) are investigated. In the ensemble control experiments, note that the pattern of responses in the $U_{zz}$ term is similar with $\bar{q}_\varphi$ (Figs. R3 and R5). This implies that changes in $\bar{q}_\varphi$ over the Arctic in the stratosphere are mainly due to the $U_{zz}$ term. The baroclinic term plays a dominant role in modulating the $\bar{q}_\varphi$ in the Arctic stratosphere. Similar results were obtained in a study from Hu et al. (2022). A reversal of $\bar{q}_\varphi$ in the upper troposphere and lower stratosphere (UTLS) over 60–70°N (Fig. R3a) leads to a reversal of the RI from November to December. In the ensemble O3clm experiments, there is no significant $\bar{q}_\varphi$ increase in the UTLS region in November, nor did the baroclinic term provide favorable conditions (Fig. R5d). This suggests that ozone-climate interaction promotes planetary wave upward by affecting the baroclinic term, which in turn induces an increase in RI.

**More detailed information please refer to the reply to the major comment #2.**

Lines 447-449: Please explain what dynamical feedback mechanisms you are referring to here.

**Response: We appreciate the reviewer's request for clarification. The dynamical feedback mechanisms are the interactions among ozone changes, wave propagation, and the BDC. They collectively influence the Arctic stratospheric dynamics. These mechanisms are summarized as follows: (1) Ozone-induced changes in wave propagation: Nathan and Cordero (2007) pointed that wave-induced ozone heating decrease wave drag in the lower stratosphere by about 25%, favoring planetary wave propagation at this altitude. Additionally, they pointed**

out that photochemically accelerated cooling due to ozone augments the Newtonian cooling and increases the wave drag by a factor of two in the upper stratosphere, which is in accordance with our finding that ozone-climate interactions enhance the upper stratospheric EP flux convergence. (2) BDC strengthening and temperature feedback: Enhanced wave activity leads to stronger downwelling in the Arctic region during early winter, which adiabatically warms the lower stratosphere. This dynamical warming, in turn, offsets the direct longwave radiative cooling effects of increased ozone. Meanwhile, the enhanced BDC associated with ozone changes would further increase rich ozone transport from middle latitudes/upper stratosphere to the Arctic lower stratosphere, leading to the positive ozone trends during early winter.

In the revised paper, we replaced "dynamical feedback mechanisms" with "ozone-circulation feedback" which has been mentioned in the preceding analysis, in order to avoid misleading. The revised text as following (see lines 641-650):

*"The ozone-climate interactions are crucial processes in modulating above-mentioned Arctic stratospheric temperature trends. Similar to earlier findings, our study highlights the role of planetary wave activity and BDC in influencing Arctic stratospheric temperature. The present study provides more detailed information on the ozone-circulation feedback processes driven by ozone-climate interactions. The ozone-circulation feedback of interest are primarily the interactions between ozone changes, wave propagation, and BDC, which regulate the dynamics of the Arctic stratosphere. Ozone-induced changes in wave propagation could modulate the vertical motions in the Arctic lower stratosphere, leading to changes in stratospheric temperature and circulation. The ozone transport associated with circulation changes could give feedback effect on polar ozone redistribution."*

Lines 456-458: You need more ensemble members to assess and reduce experimental

uncertainties

Response: Thanks for your comment. We acknowledge that using one ensemble members limits the robustness of our results, particularly in distinguishing the effects of interannual variability from long-term trends. Therefore, in the revised manuscript, we used 5 ensemble experiments (see lines 245-247):

*"Two groups of ensemble climate model experiments (i.e., the control experiment and O3clm experiment) use identical boundary conditions and initial conditions. Each group simulation consists of 5 ensemble members, with initial temperature conditions randomly perturbed."*

The analysis of new results derived from ensemble experiments can be found in Main comment #2 of the reviwer 1.

**References:**

Abalos, M., Randel, W. J., Kinnison, D. E., and Serrano, E.: Quantifying tracer transport in the tropical lower stratosphere using WACCM, Atmos. Chem. Phys., 13, 10591–10607, https://doi.org/10.5194/acp-13-10591-2013, 2013.

Albers, J. R. and Nathan, T. R.: Ozone Loss and Recovery and the Preconditioning of Upward-Propagating Planetary Wave Activity, J. Atmos. Sci., 70, 3977–3994, https://doi.org/10.1175/JAS-D-12-0259.1, 2013.

Andrews, D. G., Holton, J. R., and Leovy, C. B.: Middle atmosphere dynamics, Academic Press, Orlando, 489 pp., 1987.

Cionni, I., Eyring, V., Lamarque, J. F., Randel, W. J., Stevenson, D. S., Wu, F., Bodeker, G. E., Shepherd, T. G., Shindell, D. T., and Waugh, D. W.: Ozone database in support of CMIP5 simulations: results and corresponding radiative forcing, Atmos. Chem. Phys., 11, 11267–11292, https://doi.org/10.5194/acp-11-11267-2011, 2011.

Eyring, V., Arblaster, J. M., Cionni, I., Sedláček, J., Perlwitz, J., Young, P. J., Bekki, S., Bergmann, D., Cameron‑Smith, P., Collins, W. J., Faluvegi, G., Gottschaldt, K.‑D., Horowitz, L. W., Kinnison, D. E., Lamarque, J.‑F., Marsh, D. R., Saint‑Martin, D., Shindell, D. T., Sudo, K., Szopa, S., and Watanabe, S.: Long‑term ozone changes and associated climate impacts in CMIP5 simulations, J. Geophys. Res.-Atmos., 118, 5029 – 5060, https://doi.org/10.1002/jgrd.50316, 2013.

Hu, D., Guo, Y., and Guan, Z.: Recent Weakening in the Stratospheric Planetary Wave Intensity in Early Winter, Geophys. Res. Lett., 46, 3953–3962, https://doi.org/10.1029/2019GL082113, 2019.

Hu, Y., Tian, W., Zhang, J., Wang, T., and Xu, M.: Weakening of Antarctic stratospheric planetary wave activities in early austral spring since the early 2000s: a response to sea surface temperature trends, Atmos. Chem. Phys., 22, 1575–1600, https://doi.org/10.5194/acp-22-1575-2022, 2022.

Jones, C. D., Hughes, J. K., Bellouin, N., Hardiman, S. C., Jones, G. S., Knight, J., Liddicoat, S., O'Connor, F. M., Andres, R. J., Bell, C., Boo, K.-O., Bozzo, A., Butchart, N., Cadule, P., Corbin, K. D., Doutriaux-Boucher, M., Friedlingstein, P., Gornall, J., Gray, L., Halloran, P. R., Hurtt, G., Ingram, W. J., Lamarque, J.-F, Law, R. M., Meinshausen, M., Osprey, S., Palin, E. J., Parsons Chini, L., Raddatz, T., Sanderson, M. G., Sellar, A. A., Schurer, A., Valdes, P., Wood, N., Woodward, S., Yoshioka, M., and Zerroukat, M.: The HadGEM2-ES implementation of CMIP5 centennial simulations, Geosci. Model Dev., 4, 543–570, https://doi.org/10.5194/gmd-4-543-2011, 2011.

Simpson, I. R., Blackburn, M., and Haigh, J. D.: The Role of Eddies in Driving the Tropospheric Response to Stratospheric Heating Perturbations, J. Atmos. Sci., 66, 1347–1365, https://doi.org/10.1175/2008JAS2758.1, 2009.

Monier, E. and Weare, B. C.: Climatology and trends in the forcing of the stratospheric ozone transport, Atmos. Chem. Phys., 11, 6311–6323, https://doi.org/10.5194/acp-11-6311-2011, 2011.

Nathan, T. R. and Cordero, E. C.: An ozone-modified refractive index for vertically propagating planetary waves, J. Geophys. Res.-Atmos., 112, 2006JD007357, https://doi.org/10.1029/2006JD007357, 2007.

Zhang, J., Xie, F., Tian, W., Han, Y., Zhang, K., Qi, Y., Chipperfield, M., Feng, W., Huang, J., and Shu, J.: Influence of the Arctic Oscillation on the Vertical Distribution of Wintertime Ozone in the Stratosphere and Upper Troposphere over the Northern Hemisphere, J. Climate, 30, 2905–2919, https://doi.org/10.1175/JCLI-D-16-0651.1, 2017.

---

## Referee Report (RR1)

**Review: Effects of Ozone-Climate Interactions on the Temperature Variation**

**in the Arctic Stratosphere**

by Siyi Zhao, Jiankai Zhang, Chongyang Zhang, Zhe Wang

I appreciate the authors efforts to include more ensemble members and to adapt the ozone climatology according to my previous suggestions. I think these changes have considerably improved the results of the paper. However, I think this paper needs some restructuring and rewriting in some places before it can be presented to the audience of this journal. Specifically, I suggest:

- Removing the equations on pages 4-6. The long equations hamper the flow of the paper. The interested reader can refer to the mentioned references.
- More clearly highlighting the main pathway by which ozone trends affect temperature and circulation. In the paper, the authors make it sound as if ozone had a direct impact on the circulation and the order of the Figures is confusing. However, the actual pathway by which ozone affects the circulation is via changes of the stratospheric temperature. Therefore, I suggest moving Fig. 11 (short- and longwave heating) forward (after Fig. 4), and build the mechanistic explanation from there.
- I think a schematic outlining the proposed mechanism would help the reader to better follow the arguments made. My understanding is that in early winter, an increase of the BDC between 1980-2000 led to more ozone being transported to the pole, which decreased stratospheric temperatures during Nov-Dec due to an increase in longwave emission. This then probably leads to an increase in the BDC (mechanistic link from temperature decrease to an increase in BDC is not entirely clear to me from the manuscript), because the positive temperature difference between the control and clim_O3 run can only be due to adiabatic heating due to circulation changes (which outweighs the longwave cooling), as shown in Fig. 5.
- Are Figures 8, 9 and 10 needed for the mechanistic explanation? Otherwise move it to the supplement for better readability. I would also consider expanding Fig. 8 to include March and April.

In addition:

- In Fig. 2 it looks like in b) and c) you show twice ERA5 instead of CESM? Could this be a data error?
- Does your 1980-clim experiment use fixed SSTs? If so, which years were used to produce the SST climatology? Using only SSTs from a specific year (e.g. 1980) might skew the ozone climatology.

---

## Author Response (AR2)

**Response to Referee's Comments**

**Manuscript ID: egusphere-2024-2740**

Title: Effects of Ozone-Climate Interactions on the Long-Term Temperature Trend in the Arctic Stratosphere

Author(s): Siyi Zhao, Jiankai Zhang, Xufan Xia, Zhe Wang and Chongyang Zhang

May 2025

**Summary of revision in manuscript**

We sincerely thank the reviewer for his/her important comments and suggestions on our manuscript. The main revisions are summarized as follows:

1. We moved the formulas and equations of wave refractive index and TN wave activity flux into Supplementary Information.

2. The radiative heating analysis (the original Fig. 11) has been placed as the first figure in the mechanism explanation (now Fig. 5) to clearly connect ozone-driven temperature changes with dynamical feedbacks.

3. We added a new schematic diagram (Figure 11) to illustrate the proposed mechanism, clarifying how ozone-climate interactions drive stratospheric temperature trend changes.

**Response to Comments of Reviewer #1**

I appreciate the authors efforts to include more ensemble members and to adapt the ozone climatology according to my previous suggestions. I think these changes have considerably improved the results of the paper. However, I think this paper needs some restructuring and rewriting in some places before it can be presented to the audience of this journal. Specifically, I suggest:

- Removing the equations on pages 4-6. The long equations hamper the flow of the paper. The interested reader can refer to the mentioned references.

**Response: We appreciate the reviewers' positive feedback. Following this suggestion, the equations of refractive index and Takaya-Nakamura (T-N) wave-activity flux (the original Eqs. (4) and (10)) are moved to the Supplementary Information to ensure the readability of the paper, while retaining other key equations in the main text to support mechanistic explanations.**

- More clearly highlighting the main pathway by which ozone trends affect temperature and circulation. In the paper, the authors make it sound as if ozone had a direct impact on the circulation and the order of the Figures is confusing. However, the actual pathway by which ozone affects the circulation is via changes of the stratospheric temperature. Therefore, I suggest moving Fig. 11 (short- and longwave heating) forward (after Fig. 4), and build the mechanistic explanation from there.

**Response: Thanks for your nice suggestion. We have moved the original Figure 11 (Trends of shortwave and longwave heating rates) after Figure 4. The revised article has more clearly logic: Figures 3-4 (Temperature Trend) →Figure 5 (Radiative Feedback) →Figures 6-8 (Dynamic Feedback).**

- I think a schematic outlining the proposed mechanism would help the reader to better follow the arguments made. My understanding is that in early winter, an increase of the BDC between 1980-2000 led to more ozone being transported to the pole, which decreased stratospheric temperatures during Nov-Dec due to an increase in longwave

emission. This then probably leads to an increase in the BDC (mechanistic link from temperature decrease to an increase in BDC is not entirely clear to me from the manuscript), because the positive temperature difference between the control and clim_O3 run can only be due to adiabatic heating due to circulation changes (which outweighs the longwave cooling), as shown in Fig. 5.

**Response: Thank you for your suggestion. We have added a new conceptual diagram (Fig. 11 in the revised manuscript in Lines 587-592) that depicts the ozone–climate interaction processes as follows: "*An integrated picture depicting the mechanisms before 2000 is shown in Figure 11.*"**

[Figure]

**Figure R1 Schematic diagram of the ozone feedback mechanisms: the effects of early-winter ozone increase and late-winter/early-spring ozone decrease in the polar regions of the Northern Hemisphere on stratospheric temperature trends through ozone-climate interactions. Shown are impacts of ozone changes on radiation and dynamic, further on temperature in the lower stratosphere.**

**Its description can be seen from L402 to L413 in section Conclusion and discussion as follows:**

"*Notably, the ozone-circulation feedback of ozone-climate interactions plays a key*

*role in modulating this trend. Specifically, in early winter, ozone-circulation feedback can create an atmospheric state favorable for upward wave propagation, which is induced by the increases of $\bar{q}_{\varphi}$ in mid-latitude, and E-P flux convergence (Figs. 8, S1), which could lead to a strengthened BDC (Fig. 6) and thereby a positive trend in temperature and ozone (Figs. 3 and 7) during early winter. These trends in the BDC and planetary wave activity are predominantly driven by planetary wavenumber 1 (Figs. 6, 8). The wave-induced ozone heating increases lower-stratospheric wave propagation (Figs. 8, S1), and subsequently weakens the polar vortex during mid-winter (Fig. S1). Then, the upward propagation of planetary waves is suppressed, and consequently, the Arctic stratospheric temperature show opposite trends in January and February to early winter. During early spring, when solar radiation reaches the polar regions, reduction in ozone shortwave heating during the ozone-depletion period results in the negative temperature trend during spring (Fig. 5). After 2000, the stratospheric temperature response to ozone changes is weaker than that before 2000 (Figs. 9, 10)."*

- Are Figures 8, 9 and 10 needed for the mechanistic explanation? Otherwise move it to the supplement for better readability. I would also consider expanding Fig. 8 to include March and April.

**Response: Thank you for this helpful suggestion. The initial Fig. 8 has been relocated to Supplementary Information (Fig. S1). As the reviewers noted, Figures 9-10 are not very helpful for mechanism interpretation, so we have deleted the two figures in the revised manuscript to improve readability. The original temporal scope (Fig. S1) was retained because dynamic feedbacks dominate ozone-climate interactions in winter, while radiative feedback is the main driver in early spring.**

In addition:

- In Fig. 2 it looks like in b) and c) you show twice ERA5 instead of CESM? Could this be a data error?

**Response: We ensured that ERA5 (orange) and CESM (brown) were labeled**

**correctly. ERA5 is used to verify the reliability of the phenomenon of stratospheric temperature trends.**

- Does your 1980-clim experiment use fixed SSTs? If so, which years were used to produce the SST climatology? Using only SSTs from a specific year (e.g. 1980) might skew the ozone climatology.

**Response: Yes, the 1980-clim experiment used fixed SSTs in the year of 1980 according to the following reasons:**

**1. The year 1980 represents a period of relative stability before the rapid increase in ozone-depleting substances (ODS). At that time, stratospheric ozone had not yet experienced significant anthropogenic emitted ODS depletion (WMO, 2018).**

**2. Using fixed SSTs (prescribed as the 1980 climatology) helps isolate stratospheric dynamical responses by preventing interference from interannual variability, decadal variability or multi-decadal variability in sea surface temperatures.**

**Reference:**

**WMO: Scientific Assessment of Ozone Depletion: 2018, World Meteorological Organization Rep. 58, Geneva, Switzerland, 2018.**

---

## Author Response (AR3)

**Response to Editor's Comments**

**Manuscript ID: egusphere-2024-2740**

**Title: Effects of Ozone-Climate Interactions on the Long-Term Temperature Trend in the Arctic Stratosphere**

**Author(s):** Siyi Zhao, Jiankai Zhang, Xufan Xia, Zhe Wang and Chongyang Zhang

Jun 2025

**Summary of revision in manuscript**

We sincerely thank the editor for the important and constructive comments on our manuscript. In accordance with the suggestions, we have carefully revised the paper. The main revisions are summarized as follows:

1. We reshaped the core conclusion regarding the interaction between radiation and dynamics.

2. We have fully incorporated the valuable suggestion by introducing the perspective from McCormack et al. (2011) into the "Conclusion and discussion" section, and some sentences are toned down.

3. We updated the schematic diagram (Figure 11), reviewed and adjusted the language throughout the entire manuscript, including the abstract and main text.

**Response to Comments of Editor**

**Public justification (visible to the public if the article is accepted and published)**:
Many thanks to the authors for careful consideration and addressing of the further reviewer's concerns! I think the work is in good shape to be accepted for publication.

After careful re-reading of the manuscript, I have some final (overall minor) comments of mine regarding the interpretation of the chain of events behind the early-winter ozone climate feedback (1980-2000). In particular, I remain slightly skeptical about highlighting the role enhanced LW cooling upon higher early-winter Arctic ozone as a driver of the enhanced wave propagation and BDC (although I acknowledge that it is impossible to really identify specific drivers in these fully-coupled simulations).

First, the authors state "Lin and Ming (2021) noted that radiative damping due to longwave cooling could intensify wave dissipation and further enhance subsidence of the BDC". But the work of Lin and Ming focuses on the springtime stratosphere, when ozone absorbs SW radiation and hence any changes in ozone can either enhance or offset radiative damping rates by changing SW heating. Such a mechanism would not be at play in the winter Arctic stratosphere (as no sunlight). In addition, the enhancement of QRL under higher ozone is the result BDC strengthening, and as such it is hard to argue that it is also a driver of the BDC changes.

To me, the enhancement in BDC could just be related to just the presence of interactive ozone to begin with (before any changes in ozone take effect). For example, studies of McCormack et al. (2011, doi:10.1029/2010GL045937) showed that the presence of interactive ozone gives rise to a climatologically weaker, warmer and more disturbed winter Arctic vortex. It is possible that similar processes could also be at play when talking about Arctic trends, especially if the system leans

towards stronger BDC and warmer Arctic trend even in the absence of any ozone feedback. And once ozone is allowed to respond, increase in ozone under stronger BDC would amplify the climatological ozone-wave feedbacks, giving rise to even stronger BDC trend.

So I wonder if some sentences need to be toned down in the current manuscript, and a statement added that it is impossible to fully decouple different processes in these coupled simulations (although I agree that this study provides a great step towards that direction!). I also wonder if the left panel of figure 11 should really have "enhanced BDC/wave dissipation" at the very top? (instead of increase in O3 which could be argued is a cause of the BDC change?).

Either way, I would encourage the authors to consider the above points and see if they agree that any changes are warranted. I will then accept the manuscript for publication.

**Response: Thank you for your positive feedback and for letting us know that our manuscript is in good shape for publication.**

**We have carefully considered all the final comments on the interpretation of the early-winter ozone-climate feedback. We agree that there is inherent uncertainty in the causal relationships within a coupled model, and we have revised some statements accordingly.**

**In the main text, we revised or added some sentences as follows:**

**"*Specifically, ozone-climate interactions lead to a stratospheric state that enhances upward wave propagation and the downwelling branch of the Brewer-Dobson circulation. This leads to an adiabatic warming that significantly raises the Arctic***

*stratospheric temperature. This dynamical heating overwhelmingly offsets the longwave radiative cooling effect associated with the increased ozone during early winter."* **(in Lines 9-12).**

*"McCormack et al. (2011) pointed out that the presence of ozone-climate interactions give rise to a climatologically weaker, warmer and more disturbed polar vortex during Arctic winter"* **(in Lines 42-44).**

*"It is worth noting that this radiative effect is secondary and is overwhelmingly dominated by dynamical warming. The mechanisms behind these dominant dynamical processes will be discussed in the following analysis (Figures 6 and 8)."* **(in Lines 246-248).**

*"This dynamical heating dominates the longwave radiative cooling effect due to the ozone-climate interaction, resulting in a warming of the middle and lower Arctic stratosphere during early winter."* **(in Lines 282-283).**

*"The ozone transport associated with circulation changes could give feedback effect on polar ozone redistribution. Thus, it is plausible that the trends in stratospheric temperature and ozone are an amplification of by ozone-climate interactions. These ozone-climate interactions resemble previous findings on the climatic effects induced by zonally asymmetric ozone variations (McCormack et al. 2011; Rae et al., 2019; Zhang et al., 2020). Their experiments that account for the phase overlap between zonally asymmetric ozone heating and planetary wave centers tend to produce a climatologically warmer, weaker, and more disturbed winter Arctic vortex compared to simulations driven solely by zonal-mean ozone forcing."* **(in Lines 431-436).**

Regarding the comment on Figure 11, we fully agree that depicting the mechanism as a simple linear chain starting with an "increase in O3" could be misleading. To better represent ozone-climate interaction processes, cyclical of the process, we have completely revised the "Early Winter" panel of Figure 11.

The new schematic now illustrates the process as a feedback loop showed in Figure R1, placing the dynamical processes (enhanced BDC and upward wave propagation) at the core. This makes it clear that the system is a dynamically-driven positive feedback loop: Enhanced BDC leads to dynamic warming and increased ozone transport. Dynamic warming is the dominant factor leading to the temperature increase. Radiative cooling is a secondary response that partially offset the dynamic warming, rather than being a driver of the dynamics.

[Figure]

Figure R1. Schematic diagram of the ozone-climate interactions in the Arctic stratosphere during winter and spring. The red upward arrow indicates an increase, while the blue downward arrow denotes a decrease.